



# Socio-hydrological modeling of the tradeoff between flood
# control and hydropower provided by the Columbia
# River Treaty
Ashish Shrestha[1, *], Felipe Augusto Arguello Souza[2, *], Samuel Park[3, *], Charlotte
Cherry[4, *], Margaret Garcia[1], David J. Yu[3], Eduardo Mario Mendiondo[2]
[1] School of Sustainable Engineering and the Built Environment, Arizona State University, Tempe, AZ,
USA
[2] Department of Hydraulics and Sanitation, São Carlos School of Engineering, University of São Paulo,
São Carlos, Brazil
[3] Lyles School of Civil Engineering, Purdue University, West Lafayette, IN, USA
[4] Department of Civil and Environmental Engineering, University of Illinois at Urbana Champaign,
Urbana, IL, USA
* These authors contributed equally to this work.
*Correspondence to*: Ashish Shrestha (ashres15@asu.edu)
**Abstract.** The Columbia River Treaty (CRT) signed between the United States and
Canada in 1961 is known as one of the most successful transboundary water treaties.
Under continued cooperation, both countries equitably share collective responsibilities of
reservoir operations, and flood control and hydropower benefits from treaty dams. As the
balance of benefits is the key factor of cooperation, future cooperation could be
challenged by external social and environmental factors which were not originally
anticipated, or change in the social preferences of the two actors. To understand the
robustness of cooperation dynamics we address two research questions – i) How does
social and environmental change influence cooperation dynamics? and ii) How do social
preferences influence the probability of cooperation for both actors? We analyzed
infrastructural, hydrological, economic, social, and environmental data to inform the
development of a socio-hydrological system dynamics model. The model simulates the
dynamics of flood control and hydropower benefit sharing as a function of the probability
to cooperate, which in turn is affected by the share of benefits. The model is used to
evaluate scenarios that represent environmental and institutional change, and changes in
political characteristics based on social preferences. Our findings show that stronger
institutional capacity ensures equitable sharing of benefits over the long term. Under
current CRT, the utility of cooperation is always higher for Canada than non-cooperation
which is in contrast to the U.S. The probability to cooperate for each country is lowest
when they are self-interested but fluctuates in other social preferences scenarios.



## 1. Introduction

The Columbia River Treaty (CRT) was signed in 1961 to manage shared waters
between the United States and Canada. Under the treaty, both countries share collective
responsibilities of reservoir operations, and benefits from flood control and hydropower
production from the treaty dams equitably. CRT is known as one of the most successful
transboundary water treaties in the world, as evidenced by continued cooperation and
equitable benefit sharing (Hyde, 2010). However, since the CRT was established, external
social and environmental factors not originally anticipated, such as the degradation of
valued fish species, have affected the balance of benefits each country receives
(Bowerman et al., 2021; Trebitz and Wulfhorst, 2021). In competition and cooperation,
actors' decisions are guided by their or social preferences (also referred to as other-
regarding preferences). Actors exhibit social preferences if the actor not only cares about
their own material benefit but also cares about the material benefits of other actors (Fehr
and Fischbacher, 2002). The perceived fairness of allocated material resources or balance
of benefits, in concert with the social preferences of each actor, can significantly affect
the stability of cooperation over time (Abraham and Ramachandran, 2021; Hirshleifer,
1978; Kertzer and Rathbun, 2015; Rivera-Torres and Gerlak, 2021; Sadoff and Grey,
2002; UNESCO, 2021). Understanding these social preferences between the U.S. and
Canada helps us to understand the interplay of competition, cooperation or conflict. The
U.S. and Canada are currently renegotiating the CRT beyond 2024 with the aim of
maintaining cooperation in a changing environment. This ongoing renegotiation
motivates and raises two research questions, (1) How does social and environmental
change influence cooperation dynamics? and (2) How do social preferences influence the
probability of cooperation for both actors?

Globally, 276 transboundary river basins cover almost half of the Earth's land
surface and are the source of 60% of freshwater supplies (UN-Water, 2015; United
Nations, n.d.). Transboundary water management compounds the challenges of managing
water between competing users because the river is managed between different
jurisdictions and under different policy structures (Bernauer and Böhmelt, 2020).
Successful management of these river basins depends not only on understanding the
hydrology but also consideration of social comparison, economic needs, and political
dynamics of the upstream and downstream riparian states (Gain et al., 2021; Gober and
Wheater, 2014).  Development in transboundary river basins can result in conflict or



cooperation (Bernauer and Böhmelt, 2020). For example, the construction of dams
upstream in the Lancang-Mekong River Basin has affected the environmental conditions
and livelihood opportunities of downstream countries (Lu et al., 2021). Social factors that
can explain cooperation and conflict dynamics include asymmetric access to water
resources due to upstream-downstream locations, and varying levels of dependence on
different uses of the river (Warner and Zawahri, 2012). Transboundary rivers are
managed by multiple heterogeneous stakeholders with different sovereignty, governance
structures and economic conditions; while diverse, basin populations may be
interdependent not just hydrologically but also economically and socially (FAO, n.d.;
Rawlins, 2019). Further, the ability to sustain cooperation can be critically affected by
how benefits (e.g., water supply, hydropower) and risks (e.g., floods, droughts) are shared
under changing conditions (Wolf, 2007; Zeitoun et al., 2013). The Nile River Basin is an
example of inequitable benefit sharing where Egypt and Sudan hold absolute rights to
use, motivating conflict and international deliberation (Kameri-Mbote, 2007; Wiebe,

2001).


The history of transboundary river basins shows the challenges of cooperation in

transboundary river basins when benefits and risks are distributed inequitably. If no
agreements are in place to govern the sharing of benefits and risks, they may be
distributed according to existing levels of political or economic power or following
geographic advantages (Dombrowsky, 2009). Further, these imbalances in power can
decrease the likelihood of successfully negotiating such an agreement (Espey and
Towfique, 2004; Song and Whittington, 2004). When riparian actors cooperate, they can
achieve a wide variety of benefits, including: (1) benefits to the river; (2) benefits from
the river; (3) the reduction of costs because of the river; and (4) benefits beyond the river
(Sadoff and Grey, 2002, 2005). Examples of these benefits include flood and drought
mitigation, improved environmental conditions, and economic benefits from hydropower
or agriculture (Qaddumi, 2008).

In the case of the Columbia River, the upstream actor (Canada) operates its dams
in a way that provides a greater benefit to the downstream actor (the U.S.) in the form of
flood protection because the benefit sharing provision of the CRT ensures that Canada
receives a share of those benefits in return. The U.S. operates its dams to maximize
hydropower production and, in exchange, compensates Canada for half of the estimated
increase in hydropower benefit generated by the Treaty, which provides an economic
incentive to cooperate. This is consistent with the theory that countries tend to cooperate
when the net economic and political benefits of cooperation are greater than the benefits
from unilateral action, and when the generated benefits are shared in a way that is
perceived to be "fair" by both parties (Grey et al., 2016; Jägerskog et al., 2009; Qaddumi,
2008). The CRT was established on these grounds, as both actors agreed that the greatest
benefit of the Columbia River could be secured through cooperative management (BC
Ministry of Energy and Mines, 2013; Yu, 2008). This agreement focuses on the equitable
sharing of benefits created from cooperation, rather than on water allocation itself, which
is a key provision of some of the world's most successful water agreements (Giordano
and Wolf, 2003). The interplay of cooperation and conflict between actors can be better
understood by considering the actors' social preferences (Fehr and Fischbacher, 2002;
Kertzer and Rathbun, 2015). Behavioral economics states that decision makers have
social preferences and that the cooperating actors care about gain not only for themselves
but also for others (Kertzer and Rathbun, 2015). In general, social preferences can be
classified into four types – inequity aversion, social welfare, selfishness, and
competitiveness (Charness and Rabin, 2002). Inequity aversion is defined as actor
preferring fairness, and when benefits are evenly distributed among all group members
(Fehr and Schmidt, 1999). It is now widely accepted that humans have a strong social
preference for inequity aversion at both individual and organizational level, and that this
type of social preference is often a key to why cooperation emerges and is sustained
among unrelated individuals (Choshen-Hillel and Yaniv, 2011; Kertzer and Rathbun,
2015). Social welfare refers to actors sacrificing from their own gains to enhance the
payoffs for all group members, especially for recipients with disadvantages (Charness
and Rabin, 2002). Selfishness describes a scenario where actors only care about their own
benefits, but do not care about the payoff others receive. Finally, competitiveness assumes
that actors prefer higher payoffs than others. Understanding the social preferences
between actors (here the U.S. and Canada), could suggest how their cooperation behavior
may change, impacting the robustness of CRT.

Traditional water resource management assumes values and preferences to be
exogenous to the water resources systems, but values and preferences can co-evolve with
natural systems (Caldas et al., 2015; Sivapalan and Blöschl, 2015). Socio-hydrology, the
study of coupled human-water systems, fills this need by providing tools to represent



dynamic feedback between the hydrological and social systems (Sivapalan et al., 2012;
Troy et al., 2015). Socio-hydrological studies have explored a variety of emergent
phenomena that result from such feedback, including the levee effect, the irrigation
efficiency paradox, and the pendulum swing between human and environmental water
uses (Khan et al., 2017). In the study of transboundary rivers, socio-hydrology allows for
the explicit inclusion of changing values or preferences, and enabling assessment of
cooperation and conflict as values and preferences shift (Sivapalan and Blöschl, 2015).
Thus, we develop a socio-hydrological system dynamics model motivated by the
experience of the Columbia River to answer the research questions defined above. This
research builds upon the work of Lu et al. (2021), where the authors applied socio-
hydrological modeling to the case of the transboundary Lancang-Mekong River, by
assessing how preferences and attitudes toward cooperation affect their probability of
adhering to the agreement. The objective of this study is to quantify the balance of
benefits under cooperative reservoir operations to assess the impact of changing social
and environmental conditions as well as shifts in the social preferences of the U.S. and
Canada. While the study does not aim to provide specific recommendations for treaty re-
negotiations, it explores the role that changes in environmental priorities play in
cooperation and presents scenarios to inform future renegotiations of the CRT.

This article is organized as follows. Sect. 2 provides a general background of the
Columbia River system and treaty dams. Sect. 3 discusses the conceptualization and
formulation of the socio-hydrological model. Four scenarios based on environmental and
institutional change, and four scenarios based on behavioral economics using social
preferences are presented here. Sect. 4 explains the model testing and scenario analysis.
Sect. 5 discusses the findings of this study, draws out major conclusions gained through
this study and identifies remaining questions for future research.

## 2. Columbia River system and treaty dams

The Columbia River as depicted in Fig. 1, with its headwaters located in the
mountains of British Columbia, has a basin that extends 670,807 km$^2$ into seven U.S.
states – Washington, Oregon, Idaho, Montana, Nevada, Utah, and Wyoming – before
reaching the Pacific Ocean in Oregon (Cosens, 2012). Figure 1 also shows the location
of the treaty dams along the Columbia River. While only 15% of the river's length flows
through Canada, 38% of the average annual flow originates there (Cosens, 2012). By



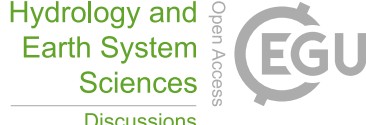

volume it is the fourth largest river in North America producing 40% of all the U.S.
hydropower, and millions of people in the Pacific Northwest (including 8 million people
in Columbia Basin (Lower Columbia Estuary Partnership, n.d.)) rely on the river for
hydropower, fishing, irrigation, recreation, navigation, and other environmental services
(White et al., 2021).

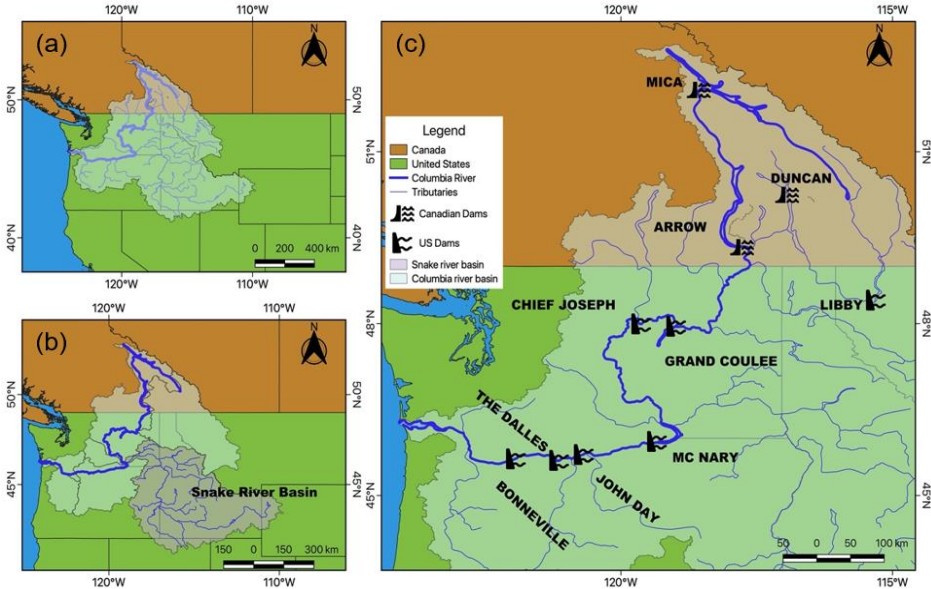


**Figure 1.** Map showing (a) the Columbia River Basin across Canada and the U.S., (b)
the Snake River Basin and its tributaries within the Columbia River Basin, and (c)
location of treaty dams along Canada and the U.S. which are also included in the socio-
hydrological system dynamics model

Hydropower development started in the Pacific Northwest in 1933 and expanded
after the CRT was established. Between 1938 and 1972, eleven dams were built on the
U.S. portion of the Columbia River, which generate over 20,000 megawatts of power (BC
Ministry of Energy and Mines, 2013). In total, there are 31 federal dams in the Columbia
River Basin that are owned and operated by the U.S. Army Corps of Engineers (USACE)
and the U.S. Bureau of Reclamation, which produce around 40 percent of electricity for
the Pacific Northwest (Bonneville Power Administration, 2001; Northwest Power and
Conservation Council, 2020c, 2020d; Stern, 2018). Dams along the Canadian side of the
Columbia River produce around half of the province's hydropower generation
(Government of British Columbia, 2019). Figure 1c shows the locations of major CRT





dams considered in the system dynamics model. The reservoir capacity of Canadian treaty
dams is 36,810 million m$^3$ of which 28,387 million m$^3$ is allocated for flood protection in
the U.S. and the capacity of the U.S. treaty dams is 11,577 million m$^3$.  Grand Coulee is
the largest and furthest upstream dam on the U.S. side. Thus, inflow to the Grand Coulee
includes the outflow from the Canadian dams and external tributaries that intersect with
the river. Flooding had been the major concern in the downstream portion of the Columbia
River. For example, the flood in Vanport, Oregon, in 1948 motivated the construction of
additional storage dams along the river (Sopinka and Pitt, 2014). This flood was the
impetus for the U.S. to seek cooperation with Canada because it was not possible to build
sufficient storage along the downstream portion of the river to protect from large floods.
The summary of dams along the Columbia River is given is Table 1.

**Table 1.** List of dams represented by the model. Projects that do not present Usable
Storage Capacity are run-off-the-river dams. Treaty Storage Commitment refers to the
room available to accommodate glacier waters under the CRT.

| Project | Reservoir formed | Country | Total Storage capacity (km$^3$) | Usable Storage capacity (km$^3$) | Treaty Storage Commitment (km$^3$) | HP Capacity (MW) | Year of Completion |
|---|---|---|---|---|---|---|---|
| Mica Dam | Kimbasket Lake | Canada | 24.7 | 14.8 | 8.6 | 1,736 | 1973 |
| Duncan Dam | Duncan Lake | Canada | 1.77 | 1.73 | 1.73 | - | 1967 |
| Keenleyside Dam | Arrow lake | Canada | 10.3 | 8.76 | 8.8 | 185 | 1968 |
| Grand Coulee | Franklin D. Roosevelt Lake | The USA | 11.6 | 6.4 | - | 6,809 | 1941 |
| Chief Joseph | Rufus Woods Lake | The USA | 0.6 | - | - | 2,069 | 1955 |
| McNary | Lake Wallula | The USA | 0.23 | - | - | 980 | 1994 |
| John Day | Lake Umatilla | The USA | 0.54 | - | - | 2,160 | 1971 |
| The Dalles | Lake Celilo | The USA | 0.41 | - | - | 2,100 | 1957 |
| Bonneville | Lake Bonneville | The USA | 0.66 | - | - | 660 | 1938 |


The original agreement during 1960s prioritized flood control and hydropower, but

emerging social and environmental concerns have shifted the way that reservoirs are
operated within the Columbia River Basin. Dam construction altered the hydrology
significantly by moderating the strong seasonal flow variability, impacting ecosystem



health. For example, changes to salmon spawning habitat, elevating smolt and adult
migration mortality and leading to declines in the salmon population (Kareiva et al.,
2000; Karpouzoglou et al., 2019; Natural Resource Council, 1996; Northwest Power
Planning Council, 1986; Williams et al., 2005). After the 1970s, mounting social
pressure to protect the aquatic environment resulted in changes in dam operations that
shifted the economic benefits that the countries receive from cooperation (Bonneville
Power Administration, 2013; Leonard et al., 2015; Northwest Power and Conservation
Council, 2020b, 2020a). This increased prioritization of ecosystem health is also seen in
other transboundary river basins (Giordano et al., 2014). With changing priorities and
operations affecting both actors' share of benefits, incentives to cooperate are shifting.

**3.  Methodology**

In this section we present the conceptual model of Columbia River system under

CRT, the formulation of a system dynamics model, model calibration and validation, and
scenario analysis. To incorporate the transboundary dynamics and feedback between the
hydrological and social systems, we simplify the representation of the hydrology and
reservoir operations by aggregating the CRT treaty dams for Canada and the U.S. To
understand the long-term dynamics of cooperation and robustness of the cooperation
under change, four scenarios based on plausible cases of environmental and institutional
change, and four scenarios based on social preferences were developed and tested as
discussed below.

### 3.1 Socio-hydrological system dynamics model

The overview of the modeling framework is illustrated with a causal loop (CL) diagram
in Fig. 2.

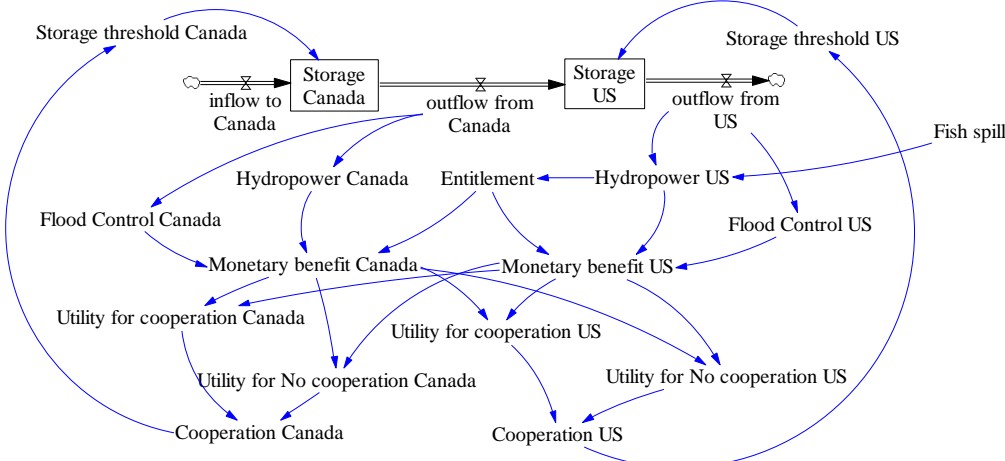

**Figure 2.** The causal loop diagram presents the hydrological and cooperation feedbacks
between the upstream and downstream countries

The storage capacity of Canada (upstream) and the U.S. (downstream) are two
important state variables which represent the aggregated storage of the treaty dams (Fig.
2). Three Canadian dams namely Mica, Duncan and Keenleyside are lumped into a single
storage as all three dams are multifunctional for flood control and hydropower
production. In the U.S., the Grand Coulee dam is the only multifunctional dam with
useable storage for flood control. These dams along the Columbia River either have
significant flood control capacity or significant hydropower production capacity (Table
1). Other hydrological components in the model (i.e., flows in the CL diagram) are inflow
into Canadian storage, outflow from Canadian storage plus intermediate tributaries,
inflow into U.S. storage, and outflow from U.S. storage. The outflow of each country's
storage is used to calculate flood control and hydropower production for each country,
which is converted into monetary units as shown in the CL diagram (Fig. 2). The U.S.
provides additional benefits to Canada through the Canadian Entitlement, a payment
equal to half of the expected additional hydropower generated due to cooperative
management of the CRT dams. Thus, the simplified reservoir operation described below



in Sect. 3.2.1 was implemented in the lumped storages on each side of the border, which
represent collective operation of all the treaty dams within each country.

The basis of the model is that each country has responsibility over operating its
own dams. Under the cooperative regime both countries operate their dams to fulfill the
requirements of the CRT. This means that Canada operates to maximize flood control
while the U.S. operates to maximize hydropower, and the benefits are shared between
both countries. As discussed in the literature (BC Ministry of Energy and Mines, 2013;
Giordano and Wolf, 2003; Grey et al., 2016; Jägerskog et al., 2009; Qaddumi, 2008; Yu,
2008), countries are expected to continue cooperating if they perceive the benefits to be
shared equitably. On the other hand, under the non-cooperative regime, the balance of
benefits is not perceived to be equitable; thus, the countries would operate their reservoirs
for their own benefit. Reservoir operation to maximize flood control and to maximize
hydropower production are in opposition for Canada and the U.S. This is because
operation for maximizing flood control requires drawdown of reservoir storage to provide
space for incoming high flows, while operation for maximizing hydropower production
requires reservoir storage to be maintained at higher levels to achieve the highest
hydraulic head possible. In a non-cooperative regime, Canada would likely switch
operations to maximize hydropower production while the U.S. would have to decrease
storage or water level to provide flood control, at the detriment of U.S. hydropower
production.

### 3.2 Equations and parameters
Equations describing the links between stocks and flow variables as shown in the
CL diagram (Fig. 2) are categorized into reservoir operation, cooperation dynamics,
economic benefits, and environmental spills. These equations mathematically describe
hydrological processes, as well as feedback from social and economic variables. The
following sections describe the formulation of equations for each part of the system in
greater detail. The inflow, outflow, water level and storage data are presented in Fig. S2–
S10, supplemental material (SI 1).

### 3.2.1   Reservoir operation
The monthly change in Canadian and the U.S. storage (m³ month$^{-1}$) as the function
of inflow and outflow is given in Eq. (1) and (2).



$$\frac{dS_{CA}}{dt} = Q_{i_{CA}} - Q_{o_{CA}} \tag{1}$$

$$\frac{dS_{US}}{dt} = Q_{i_{US}} - Q_{o_{US}} \tag{2}$$

The Canadian inflow ($Q_{i_{CA}}$) corresponds to the streamflow observed upstream of Mica
and Duncan dams and the difference between Mica outflow and Arrow inflow (i.e. flow
from intermediate tributaries). The data was retrieved from the Bonneville Power
Administration (Bonneville Power Administration, 2020). The U.S. inflow ($Q_{i_{US}}$) is
equal to the outflow from Canadian storage ($Q_{o_{CA}}$) plus the tributaries between the outlet
of Duncan and Arrow dams and inlet of the Grand Coulee reservoir. The flow from
tributaries on the Canadian side were calculated as the difference between the streamflow
at the International Border and outflow from Duncan and Arrow dams, while the
tributaries between the International Border and the Grand Coulee reservoir were
estimated by a linear regression (Fig. S12).
The regulated Canadian ($Q_{o_{CA}}$) and U.S. ($Q_{o_{US}}$) outflows were simulated using Eq. (3)
and (4).

$$Q_{o_{CA}} = \begin{cases} \begin{cases} Q_{CA_{max}}, for \ n_{CA} * Q_{i_{CA}} \geq Q_{CA_{max}} \\ n_{CA} * Q_{CA_{max}} + max\left[0, min\left(Q_{CA_{max}} - n_{CA} * Q_{i_{CA}}, \frac{S_{CA} - S_{CA_{threshold}}}{2592000}\right)\right], \end{cases}, (for \ I_1) \\ \begin{cases} Q_{CA_{max}}, for \ Q_{i_{CA}} \geq Q_{CA_{max}} \\ Q_{i_{CA}} + max\left[0, min\left(Q_{CA_{max}} - Q_{i_{CA}}, \frac{S_{CA} - S_{CA_{threshold}}}{2592000}\right)\right], \end{cases}, (otherwise) \end{cases} \tag{3}$$

where $I_1$ is the condition when $S_{CA} + Q_{i_{CA}} * 2592000 < S_{CA_{threshold}}$, and

$n_{CA}$ parameter maintains the dynamic storage threshold required for flood control.

$$Q_{o_{US}} = \begin{cases} \begin{cases} Q_{i_{US}}, for \ Q_{i_{US}} \geq Q_{US_{max}} \\ Q_{i_{US}} + max\left[0, min\left(Q_{US_{max}} - Q_{i_{US}}, \frac{S_{US} - S_{US_{threshold}}}{2592000}\right)\right], \end{cases}, (for \ I_2) \\ Q_{i_{US}} + \frac{S_{US} - S_{US_{threshold}}}{2592000}, otherwise \end{cases} \tag{4}$$

where $I_2$ is the condition when $S_{US} + Q_{i_{US}} * 2592000 < S_{US_{max}}$.


Outflow was computed as a dependent variable of:

309    a) inflows ($Q_{i_{CA}}$ and $Q_{i_{US}}$),



b)  maximum outflows observed in the Canadian side (Arrow and Duncan

dams - $Q_{CA_{max}}$), and in the U.S. side (Grand Coulee - $Q_{US_{max}}$),

c)  the maximum storage capacity of Canadian lumped dam ($S_{CA_{max}}$) and the

Grand Coulee dam ($S_{US_{max}}$),

d)  the updated storage stage at each time step in the lumped Canadian

reservoir and the Grand Coulee reservoir ($S_{CA}, S_{US}$) and

e)  the dynamic storage threshold for each side ($S_{CA_{threshold}}, S_{US_{threshold}}$)

The dynamic storage thresholds (m$^3$) variable, mentioned in Eq. (3) and (4), was
estimated according to the simplified reservoir operation given by Eq. (5) and (6) and is
schematically represented by Fig. 3. It determines the operational level of the reservoirs
based on the probability of cooperation (i.e., the higher the cooperation, higher coherence
with the CRT agreement).

$$S_{CA_{threshold}} = S_{CA_{FC}} * C_{CA} + (1 - C_{CA}) * S_{CA_{HP}} \qquad (5)$$

$$S_{US_{threshold}} = S_{US_{HP}} * C_{US} + (1 - C_{CA}) * S_{US_{FC}} \qquad (6)$$

As explained above, we consider two operation schemes for each country: (1) operate to
maximize for flood control or (2) operate to maximize for hydropower production.
Depending on the state of cooperation, the choice will change. In most cases, the system
will depend on what Canada chooses, and the U.S. will have to alter its operations in
response. Therefore, when the Canadian probability to cooperate parameter ($C_{CA}$)
approaches one, Canada is fully cooperating. Under cooperation, we assume that Canada
operates to maximize flood control and the U.S. operates to maximize hydropower.
Conversely, when $C_{CA}$ approaches zero, this would indicate lack of cooperation. Under
non-cooperation, the Canadian side does not provide flood storage to the U.S. and, after
a few simulation time steps where the U.S. endures higher flood damages, the U.S.
switches from the hydropower production regime ($S_{US_{HP}}$) to the flood control regime to
optimize its benefits ($S_{US_{FC}}$). The target flood control storage in Canada ($S_{CA_{FC}}$) was
determined based on average historical storage in the three treaty reservoirs, while the
hypothetical hydropower scheme was assumed as the dams operating at 95% of their full
production capacity. The U.S. monthly target storages under the hydropower scheme
($S_{US_{HP}}$) were determined based on the historical monthly average, while the hypothetical
target storage to provide themselves protection against floods was calculated as the
additional room that Canada would not provide in case of switching to the hydropower



scheme $S_{CA_{HP}}$ as presented in Eq. (5) and (6). Therefore, the storage will be dependent on
cooperation. The probability to cooperate variables $C_{CA}$ and $C_{US}$ are described in the Sect.

3.2.2.

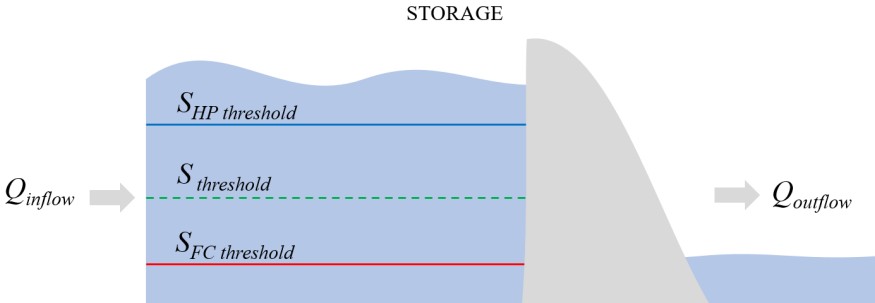


**Figure 3.** Schematic representation of the dynamic storage threshold ($S_{threshold}$),
represented by the green line. $S_{threshold}$ can range between the blue line, that represents
the target storage to optimize hydropower production ($S_{HP_{threshold}}$), and the red line,
that represents the target storage to avoid flood damages downstream the dam
($S_{FC_{threshold}}$)

### 3.2.2  *Cooperation dynamics*
Cooperation amongst the two actors both impacts and is impacted by reservoir
operations and benefit sharing. Unequal distribution of benefits alters the sense of fairness
and reciprocity. To conceptualize and understand the cooperation dynamics between two
actors in the context of CRT, the theory of social preferences is drawn from the field of
behavioral economics. Social preferences refer to the behavior of actors (where here
actors are countries not individuals) depending not only on their own material payoffs but
also about the material benefits of other actors (Fehr and Fischbacher, 2002). These
preferences are formalized as the utility function $u_i$, represented by Eq. (7),

$$u_i = w_i - \alpha_i * \max(w_i - w_j, 0) + \beta_i * \max(w_j - w_i, 0) \qquad (7)$$

where $w_i$ is actor $i$'s expected wealth, and $w_j$ is actor $j$'s expected wealth. The
value for $\alpha$ represents disutility from having more than the other actor (the guilt
coefficient), and $\beta$ represents disutility from having less than the other actor (the jealousy
coefficient). Among the four types of social preferences described in Sect. 1, this model
uses inequity aversion for the behavioral model of Canada and the U.S. because the
balance of benefits (Bankes, 2017; Shurts and Paisley, 2019) between these two countries
is believed to be a key factor to explain the level of cooperation.



The utility function is composed of two parts: utility from each actor's own

monetary benefits and from the other's monetary benefits. We defined the utility function
$U$ of each country in Eq. (8–11),

$$U_{CA} = w_{CA} - \alpha_{CA} * \max(w_{CA} - w_{US}, 0) + \beta_{CA} * \max(w_{US} - w_{CA}, 0) \tag{8}$$

$$U_{US} = w_{US} - \alpha_{US} * \max(w_{US} - w_{CA}, 0) + \beta_{US} * \max(w_{CA} - w_{US}, 0) \tag{9}$$

$$w_{CA} = \omega * (HP_{CA} + FC_{CA} + E) \tag{10}$$

$$w_{US} = \omega * (HP_{US} + FC_{US} - E) \tag{11}$$

where $w$ of each country is the utility from monetary benefits, $HP$ of each country is the
hydropower benefit, $FC$ of each country is the benefit from flood prevention, $E$ is the
Canadian entitlement, and $\omega$ is the coefficient that can convert the monetary values to
utility. Therefore, the sum of the second term ($\alpha$) and the third term ($\beta$) in Eq. (8) and (9)
represents the utility from the other country's monetary benefits because the country has
inequity aversion.

We use logit dynamics functions to capture the rate of change of cooperation

probability (Iwasa et al., 2010), represented by Eq. (12) and (13),

$$\frac{dC_{CA}}{dt} = \chi \left[ \frac{e^{\gamma*E[U_{CA\_coop}]}}{e^{\gamma*E[U_{CA\_coop}]} + e^{\gamma*E[U_{CA\_NoCoop}]}} - C_{CA} \right] \tag{12}$$

$$\frac{dC_{US}}{dt} = \chi \left[ \frac{e^{\gamma*E[U_{US\_coop}]}}{e^{\gamma*E[U_{US\_coop}]} + e^{\gamma*E[U_{US\_NoCoop}]}} - C_{US} \right] \tag{13}$$

where $C_{CA}$ and $C_{US}$ represent the probability of each country to cooperate (ranging from
0 for Non-Cooperation to 1 for Full Cooperation), and the probability $\chi$ if each country
is given an opportunity to choose between two strategies, independent of their last choice.
With stronger institutions or governance, $\chi$ is higher (i.e., > 0.5), with weaker institutions,
$\chi$ is lower (i.e., < 0.5). $E[x]$ stands for the expected value and $\gamma$ describes the sensitivity
of cooperation changes to the differences between expected utility values. A large $\gamma$
represents a deterministic model that actors always choose the option with the higher
expected utility value. On contrary, a small $\gamma$ indicates that the actor is likely to switch
their strategy randomly at each time step, independent of the expected utility difference.
We assumed $\gamma$ to be large and constant as both actors aims for higher expected utility.
For probability to cooperate, if $C_{CA}$ equals to 0.9 that means there is 90% likelihood that
Canada will cooperate with the U.S. and 10% likelihood it will not cooperate. Low values
of $\chi$ indicate the policy of the country over whether to cooperate or not would be less





sensitive to the current probability to cooperate and the expected utility (Hofbauer and
Sigmund, 2003).

Actors are willing to cooperate if they are confident that the other actor involved
in the cooperation problem will also cooperate; this is the basis for cooperative outcomes
as demonstrated in the context of social dilemma situation like prisoner's dilemma by
Fehr and Fischbacher (2002). A mixed strategy prisoner's dilemma is used to calculate
the expected monetary payoffs, *E[w]*, according to the combination of strategic decisions
across countries (Table 2). For example, $w_{CA_{CN}}$ is the monetary benefit of Canada when
the U.S. chooses to cooperate and Canada chooses to not cooperate. In this case, the
expected utility of Canada from monetary benefits is calculated by Eq. (14). Similar,
equation not shown here was used for the U.S. to calculate its expected utility.
Afterwards, the expected utility of Canada is calculated involving disutility of inequity
aversion using Eq. (15) and (16), and similar equations not shown here was used for the
U.S.

$$E[w_{CA}] = E\left[w_{CA_{Coop}}\right] * C_{CA} + E\left[w_{CA_{NoCoop}}\right] * (1 - C_{CA}) \tag{14}$$

$$E\left[U_{CA_{coop}}\right] = E\left[w_{CA_{Coop}}\right] - \alpha_{CA} * \max\left(E\left[w_{CA_{Coop}}\right] - E[w_{US}], 0\right)$$
$$+ \beta_{CA} * \max(E[w_{US}] - E[w_{CA\_Coop}], 0) \tag{15}$$

$$E\left[U_{CA_{nocoop}}\right] = E\left[w_{CA_{NoCoop}}\right] - \alpha_{CA} * \max\left(E\left[w_{CA_{NoCoop}}\right] - E[w_{US}], 0\right)$$
$$+ \beta_{CA} * \max(E[w_{US}] - E[w_{CA\_NoCoop}], 0) \tag{16}$$


**Table 2.** The payoff matrix of the mixed strategy prisoner's dilemma between Canada
and U.S. showing monetary benefit for Canada ($w_{CA\_}$) and the U.S. ($w_{US\_}$) in four
conditions: *CC* – the U.S. and Canada both cooperate, *CN* - the U.S. cooperate and
Canada do not, *NC* - the U.S. do not cooperate and Canada do, and *NN* – the U.S. and
Canada both do not cooperate

| Canada<br>US | Coop<br>$(C_{CA})$ | No Coop<br>$(1 - C_{CA})$ |
|---|---|---|
| Coop<br>$(C_{US})$ | $(w_{US_{CC}}, w_{CA_{CC}})$ | $(w_{US_{CN}}, w_{CA_{CN}})$ |
| No Coop<br>$(1 - C_{US})$ | $(w_{US_{NC}}, w_{CA_{NC}})$ | $(w_{US_{NN}}, w_{CA_{NN}})$ |






### 3.2.3    *Economic benefit equations*

The model simulates the benefits that both countries receive from the river. The default
operation assumes that the countries cooperate to maximize benefits across the whole
system, while in the counter case benefits are based on operation of each side individually.
The economic benefits related to flood control are accounted as the damages prevented
by the reservoir storage operations. Although the U.S. Corps of Engineers reports that
flood damages in Trail, British Columbia, a city near the International Border, occur when
streamflow exceeds 6,371 $m^3$ $s^{-1}$ (225,000 cfs) (USACE, 2003), we did not find details
about the damages related to the seasonal flows in Canada. Therefore, the associated
economic benefit due to the damages prevented for the Canadian side due to reservoir
operation was assumed to be negligible.

In the U.S., significant damages occur when streamflow exceeds 12,742 $m^3$ $s^{-1}$ at

Dalles, Oregon, and major damages are caused when flows reach 16,990 $m^3$ $s^{-1}$ (Bankes,
2012). Therefore, when they are operating jointly, Canada must draw down storage
reservoirs before April 1 to accommodate spring runoff and avoid peak flows
downstream. Otherwise, we assume that the U.S. must switch to a flood control scheme.
Flood damages prevented because of reservoir management under CRT were explored by
Sopinka and Pitt (2014). They compared the maximum annual daily peak flows at Dalles
after the implementation of the CRT, and the corresponding monetary damages they
could have caused without flood control storage provided. The results of their study were
fitted to an exponential curve using Eq. (17) which gives economic benefit in the U.S.
due to flood control,

$$FC_{US} = 4.007 * exp^{(2*10^{-4}*Q_{Dalles})} \tag{17}$$

which presented a R-squared value equal to 0.76. This function was used to estimate the
value of flood protection. More details on flood control benefit are presented in Fig. S11–
S13, supplementary material (SI 2).

The economic benefit in the U.S. due to flood damages avoided ($FC_{US}$ ) is based

on inflow ($m^3$ $s^{-1}$) into the Dalles dam ($Q_{Dalles}$). Thereafter, we found the correlation
between the Dalles's inflow and the combined outflow of Grand Coulee ($Q_{Grand\ Coulee}$)
and the Snake River ($Q_{Snake\ River}$) (Eq. 18).





$$Q_{Dalles} = 1.132 * (Q_{Grand\ Coulee} + Q_{Snake\ River}) + 0.0137 \qquad (18)$$

The Snake River discharge was included in this analysis because its basin is the major
tributary to the Columbia River, contributing to flow at the Dalles.

The other economic benefit resulting from management of the Columbia River is
the electricity produced by the hydropower facilities installed in the dams listed in Table
1. Although other dams on the Canadian side of the Columbia Basin have capacity to
generate hydropower, the model only considers those three that are part of the CRT.
Similarly, we only consider the six federal dams on the U.S. side whose surplus
production contributes to the determination of the Canadian Entitlement. Since all six
dams produce energy but only the Grand Coulee operations were modeled, we split the
economic benefit from hydropower generation in two parts. Equation 19 resulted from
the regression performed between the product of the forebay level ($h$) times Grand
Coulee's monthly average outflow ($Q_{out}$) versus the average monthly historical
hydropower produced by Grand Coulee ($HP_{Grand\ Coulee}$) (MWh), which resulted in an
R-squared equal to 0.89.

$$HP_{Grand\ Coulee} = 1.2797(Q_{out} * h) + 288616 \qquad (19)$$


In addition, we calculated the electricity produced by the other five dams in Eq.

(20):

$$HP_{5\ dams} = \begin{cases} 1208.9 * (W_{fish} * Q_{out}) \ for \ W_{fish} * Q_{out} \le 400 m^3 s^{-1} \\ 833.9 * (W_{fish} * Q_{out}) \ for \ W_{fish} * Q_{out} > 400 m^3 s^{-1} \end{cases} \qquad (20)$$

where $HP_{5\ dams}$ is the hydropower in MWh produced by Chief Joseph, McNary, John
Day, the Dalles and Bonneville dams. The variable $Q_{out}$ is Grand Coulee's monthly
outflow and $W_{fish}$ is the weighting factor that considers the operations to meet
environmental demands, which is detailed in Sect. 3.2.4. The correlation for the first and
second conditions in Eq. (20) presented R-squared values equal to 0.99 and 0.94,
respectively. Correlation to predict hydropower generation from outflows and forebay
levels are presented in Fig. S14–S15, supplementary material (SI 2). In Eq. (21) we
calculate the total economic benefit due to hydropower production ($HP_{US}$) in USD,

$$HP_{US} = (HP_{Grand\ Coulee} + HP_{5\ dams}) * HP\$_{US} \qquad (21)$$

where $HP\$_{US}$ is the average energy price of Oregon and Washington states according to
the (U.S. Energy Information Administration, n.d.).

For the Canadian dams, historical data on hydropower production is not available.
Therefore, Eq. (22) estimates the economic benefit due to electricity produced in Canada
($HP_{CA}$) in USD based on the generation flow capacity ($Q_{turb}$), the maximum hydraulic
head ($H$), the hydropower facility efficiency ($\mu$), the specific water weight ($\gamma$) and the
electricity price in British Columbia according to (BC Hydro, n.d.).

$$HP_{CA} = \frac{\mu * \gamma * Q_{turb} * H}{10^3} * HP\$_{CA} \tag{22}$$

Since this equation is based on the Mica dam and, in the model, the three Canadian dams
are modeled together, the $Q_{turb}$ and $H$ were interpolated according to the actual and
maximum recorded Canadian outflow and Canadian storage, respectively.

The last economic benefit modeled in this study is the entitlement that U.S. returns
to Canada as a payment for increased hydropower generation due to the collaboration
between both countries. The Canadian Entitlement ($E$) simulated in USD is a function of
the actual Entitlement in MWh provided by the U.S., the $\kappa$ parameter, which corresponds
to a dimensionless correction factor of the total energy produced by the US, and the
average energy price $HP\$_{US}$ of Oregon and Washington states (Eq. 23).

$$E = Entitlement * \kappa * HP\$_{US} \tag{23}$$


**3.2.4     Impact of environmental spills**
The Fish Operation Plan (FOP) details the spills dams must release to meet
biological requirements. Fish passage facilities have decreased hydropower generation
(Northwest Power and Conservation Council, n.d.). The Bonneville Power
Administration, which operates the U.S. treaty dams, estimates that loses due to forgone
revenue and power purchases are about $27 million to $595 million per year (Northwest
Power and Conservation Council, 2019). Although the historical data between 1985 and
2018 of hydropower generated by the 6 U.S. dams listed in Table 1 reveal hydropower
production increased after the FOP implementation, when normalized as the ratio of
hydropower production to inflows, there is in fact a decrease in production after FOP is
implemented.

In order to address the impact of biological spills on hydropower production, we
created a weighting factor in the hydropower benefit equation for the U.S., which is
detailed in Eq. (24).





$$W_{fish} = \frac{\sum_{i=1}^{5} \frac{Q_{fish_i}}{Q_{outflow_i}} * MaxHP_i}{\sum_{i=1}^{5} MaxHP_i} \qquad (24)$$

This weighting factor ($W_{fish}$) accounts for the fraction of flow ($\frac{Q_{fish_i}}{Q_{outflow_i}}$) that no longer
goes through the hydropower turbines between April and August because it is released
through a spillway or a regulating outlet to meet the biological demands. We calculated
the average monthly fraction for each of the $i$ dams downstream of Grand Coulee and
multiplied it by the maximum hydropower produced by each dam ($MaxHP_i$) to address
individual contributions and the particular effect of FOPs at treaty dams.

***3.3 Model setup and testing***

The equations described above are formulated into the system dynamics model

and implemented in R, a statistical programming environment. In this study we used the
library package *deSolve* Version 1.28 (Soetaert et al., 2010, 2020) to solve the initial value
problem of ordinary differential equations (ODE), differential algebraic equations and
partial differential equations. The ordinary differential equations wrapper (i.e., *lsoda*) that
uses variable-step, variable-order backward differentiation formula to solve stiff
problems or Adams methods to solve non-stiff problems (Soetaert et al., 2010) was used
to compute dynamic behavior of the lumped reservoir system, and to assess how the
reservoir level and operation rules change as a function of time and different variables.
The model was simulated using monthly time steps. Sensitivity analysis was conducted
to test the sensitivity of the parameters and identify the parameters that are most
important. However, all unknown parameters were used in calibration due to the limited
computational cost. The details of the sensitivity analysis are presented in supplementary
material (SI 3).

***3.3.1 Calibration and validation***

The calibration and selection of appropriate parameter values are essential to

accurately reproduce the system's behavior. The calibration parameters can be found in
Fig. 4. These parameters are related to both the hydrological and socio-economic
components of the system. A genetic algorithm (GA) (Scrucca, 2021) was used to
optimize the system dynamics model, using observation for the period from 1990 to 2005.
The methodological framework for model calibration is presented in Fig. 4. A single


objective function was defined as minimizing the average root mean square error of
reservoir water levels in Canada and the U.S. (Z), which is given by Eq. (25).

$$Z = \frac{RMSE_{Sca} + RMSE_{Sus}}{2} \tag{25}$$

A maximum of 200 iterations and a population size of 200 were used to run the algorithm
with a stopping criteria of 70 iterations before the algorithm stops when no further
improvement can be found. The selected larger population size and iterations, for eight
parameters, ensures that search space is not restricted. The range of parameter values
assigned was, 0.01 to 0.8 for $\chi$, 0.95 to 1.05 for $W_{fish}$, 0.1 to 0.5 for $n_{CA}$, 0.95 to 1.05 for
$\kappa$, 0 to 1.3 for $\alpha_{US}$ and $\alpha_{CA}$, -4 to -0.01 for $\beta_{US}$ and $\beta_{CA}$. The model was calibrated using
monthly time series data from 1990 to 2005, and fitted parameters were used to validate
the model using data from 2006 to 2017.

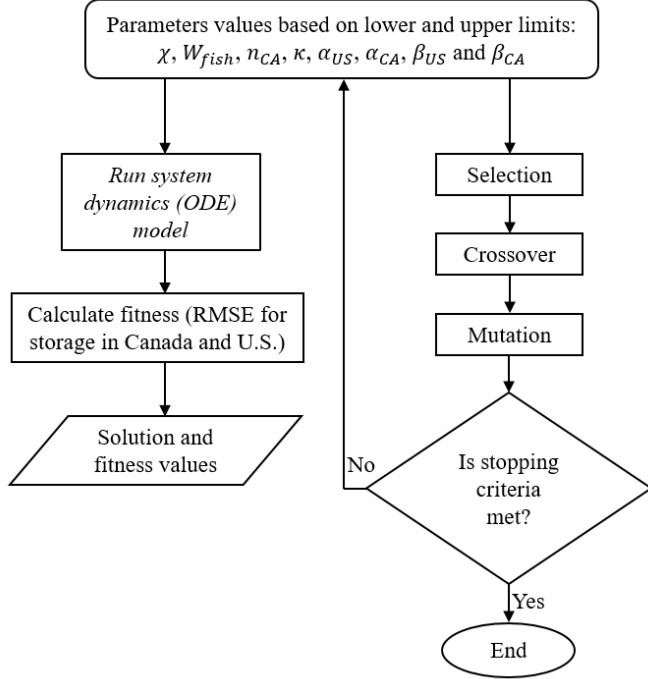


**Figure 4.** Overview of calibration process to optimize parameters values using genetic
algorithm. The stopping criteria includes either the maximum iteration for algorithm to
run which is set at 200 generations, or number of iterations before algorithm stop incase
no further optimal fitness value can be found, which is set at 70 generations

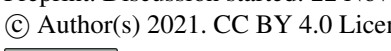

The model assessment for the goodness-of-fit between modeled and observed
values was done using four goodness-of-fit metrics, including root mean square error
(RMSE), percent bias (PBIAS), volumetric efficiency (VE) and relative index of
agreement (rd). RMSE gives the standard deviation of the model prediction error, with
lower RMSE indicating better fitness. PBIAS measures average tendency of the simulated
values to be higher or lower than the observed data, which range from $-\infty$ to $+\infty$, and its
optimal value being 0. VE is a modified form of mean absolute error in which absolute
deviation is normalized by total sum of observed data, which could range from 0 to 1,
with 1 indicating better agreement. Lastly, rd measures the agreement between simulated
and observed data, with its values ranging from $-\infty$ to 1, and 1 indicating better fit. For
mathematical expressions of these metrics readers are referred to Zambrano-Bigiarini

(2012).


*3.4 Scenario analysis*
Scenario analysis explores dynamics within cooperation and benefit sharing as a result of
external environmental factors, institutional capacity, and social and behavioral
preferences.

*3.4.1    Scenarios based on environmental and institutional change*
The CRT's success has been based on benefit sharing between the two countries (Hyde
2010). However, due to increased environmental flows in the U.S., some parties feel
benefits are no longer equitable. Based on these issues, four scenarios were developed to
represent the changes in institutional capacity and environmental factors that could affect
the probability of cooperation. The model was used to simulate the probability of
cooperation under these scenarios for 28 years between 1990 to 2017, which was
compared with the baseline scenario that represents the existing system obtained from
calibrated model. These scenarios are:
i.    *Chi ($\chi$) decreases* – The calibrated value of 0.5 decreases to 0.05. $\chi$ represents the

institutional capacity which determines the growth potential of the probability of

cooperation. This type of condition could occur due to a more tense relationship

between the U.S. and Canada that could arise due to lack of cooperation in other

areas or weaker institutions.



ii. *Chi ($\chi$) increases* – The calibrated value of 0.5 increases to 0.7. This scenario represents the strengthening of institutions. Note: The selection of $\chi$ values for scenarios "*Chi ($\chi$) increases*" and "*Chi ($\chi$) decreases*" was done based on experimentation where drastic change in $C_{ca}$ and $C_{us}$ is observed at both ends of increasing and decreasing $\chi$ from calibrated value.

iii. *High fish spills* – Environmental concerns result in prioritization of spills for fish passage. Water for fish spills increases by 40% from April through August.

iv. *Chi ($\chi$) decreases and high fish spills* – Chi ($\chi$) decreases to 0.05 and fish spills increases by 40%. It represents the scenario when environmental pressure is high, and institutions are weaker.

### 3.4.2 Scenarios based on social preferences

As discussed by Fehr and Fischbacher (2002) and Kertzer and Rathbun (2015), consideration of social preferences is required to understand mechanisms of cooperation and the effect of material or benefit payoffs. The key assumption in economic science that economic reasoning is mostly based on self-interest or that all actors are exclusively motivated by their material self-interest is invalid as this assumption rules out the heterogeneity arising from social preferences which substantial fraction of people exhibit (Fehr and Fischbacher, 2002). To explore the effect of inequality aversion of each country on the cooperation dynamics, we develop four scenarios with different configuration of $\alpha$ and $\beta$ values for Canada and the U.S. (shown in Table 3). Theoretically, the value of the two coefficients should range from $\beta < 0 < \alpha \leq 1$, and jealousy is more likely than guilt ($|\beta| > |\alpha|$) (Fehr and Schmidt, 1999). The four scenarios are:

i. *Scenario 0* – we posit that both Canada and the U.S. have the same inequality aversion ($\alpha_{ca} = \alpha_{us} = 0.9$, $\beta_{ca} = \beta_{us} = -1$). Same inequality aversion means that the actors prefer the benefits to be equally distributed i.e., each actor wants to increase/decrease their benefits up-to the equitable benchmark when there is imbalance in benefits. This scenario is not the same as the "baseline" scenario discussed above in Sect. 3.4.1, where four scenarios based on environmental and institutional change are compared.

ii. *Scenario 1* – the U.S. has less guilt than Canada ($\alpha_{ca} = 0.9$, $\alpha_{us} = 0.3$, $\beta_{ca} = \beta_{us} = -1$). That means the U.S. is willing to have more benefits than Canada.





iii.    *Scenario 2* – Canada has more jealousy than the U.S. ($\alpha_{ca} = \alpha_{us} = 0.9$, $\beta_{ca} = -3$,
$\beta_{us} = -1$). This means Canada is unwilling to have less benefits than the U.S.
iv.    *Scenario 3* – we assume that the both countries have no social preferences ($\alpha_{ca} =$
$\alpha_{us} = \beta_{ca} = \beta_{us} = 0$), which signifies self-interest or selfishness. In this scenario,
each country is only concerned with its own utility and indifferent to the utility of
the other.

We did not include the change of the jealousy of the U.S. or the guilt of Canada
in the scenario analysis. This choice is justified because the net monetary benefit of the
U.S. is always higher than that of Canada, so the U.S. never feels jealousy nor does
Canada feel guilt. In each scenario, we impose a small amount of white noise to each
country's α and β values which introduces an element of stochasticity.

**Table 3.** The configuration of different other-regarding preferences of Canada and the
U.S. for scenario analysis. In the scenario 0 both countries have the same level of
inequality aversion, while in scenario 1 the U.S. has less guilt than the scenario 0, in
scenario 2 Canada is more jealous than in the scenario 0, and in scenario 3 both countries
are only concerned with their own utility.

|  | $\alpha_{ca}$ | $\alpha_{us}$ | $\beta_{ca}$ | $\beta_{us}$ |
|---|---|---|---|---|
| **Scenario 0** | 0.9 | 0.9 | -1 | -1 |
| **Scenario 1** | 0.9 | 0.3 | -1 | -1 |
| **Scenario 2** | 0.9 | 0.9 | -3 | -1 |
| **Scenario 3** | 0 | 0 | 0 | 0 |


## 4    Results
This section presents results of model parameterization using genetic algorithm
including results from the sensitivity analysis, and results from the scenario analysis.

### 4.1 System dynamics model parameterization and testing
During the calibration period from 1990 to 2005 (and to the present) Canada and
the U.S. have conformed to the treaty, irrespective of changes in benefit sharing and
probability to cooperate. The selection of these social, economic and behavioral
parameters therefore represents conditions of cooperation regime. Based on the objective
function, the goal was to calibrate the model to simulate reservoir levels that match past
observations. Figure 5a–d shows the simulated and observed time series, during 1990 to
2005, of the stock (storages) and flow (outflow) variables along with the economic





variable of hydropower benefits for the U.S. The model performance metrics for the
calibration period are shown in Table 4. The metrics show good calibration results with
respect to all four metrics. The root mean square error and percent bias are minimal and
volumetric efficiency is higher, for both stock and flow variables. Although the
magnitude of the RMSE is large, it is considered a good fit when compared proportionally
with reservoir volumes, streamflow, and benefits.

As seen in Fig. 5a–b, the total reservoir capacity in the Canadian treaty dams far
exceeds the capacity of the U.S. treaty dams and it is to be noted that the treaty flood
control (FC) level in the Canadian dams is 28,387 million m$^3$ (equivalent to the 8.95 MAF
flood storage requested by U.S.). Grand Coulee inflow is the primary input to the U.S.
storage. Thus, the observed and computed inflows are compared to ensure accurate model
behavior (Fig. 5c). The hydropower benefit for Canada depends on U.S. hydropower
production due to the Entitlement; thus, only the benefit of the U.S. was selected for
assessing the calibration results, as estimating hydropower benefit of the U.S. correctly is
an important process in the model (Fig. 5d). Here, the Canadian Entitlement provided in
terms of energy supply is converted into monetary units to compare hydropower with
other benefits. The simulated hydropower production for the U.S. is compared to the
observed cumulative energy production data retrieved from the U.S. Army Corps of
Engineers database. The benefit in terms of the monetary value is obtained by multiplying
the average unit cost ($ MWh$^{-1}$) of energy by the hydropower quantity (MWh).

**Table 4.** Calibration (1990-2005) and validation (2006-2017) result

| Stock and flow variables | Metric | Calibration | Validation |
|---|---|---|---|
| Storage Canada | RMSE | 6844.14 Million m$^3$ | 5596.153 Million m$^3$ |
| | PBIAS (%) | 14.70 | 6.50 |
| | VE | 0.76 | 0.82 |
| | rd | 0.30 | 0.51 |
| Storage US | RMSE | 1682.46 Million m$^3$ | 1373.34 Million m$^3$ |
| | PBIAS (%) | -8.60 | -6.90 |
| | VE | 0.88 | 0.91 |
| | rd | 0.68 | 0.78 |
| GCL inflow | RMSE | 963.20 m$^3$ s$^{-1}$ | 886.23 m$^3$ s$^{-1}$ |
| | PBIAS (%) | 1.70 | 2.4 |
| | VE | 0.72 | 0.75 |
| | rd | 0.82 | 0.89 |
| HP benefit | RMSE | 144.24 Million US$ | 139.66 Million US$ |
| | PBIAS (%) | 11.30 | 15.10 |
| | VE | - | - |
| | rd | 0.66 | 0.73 |




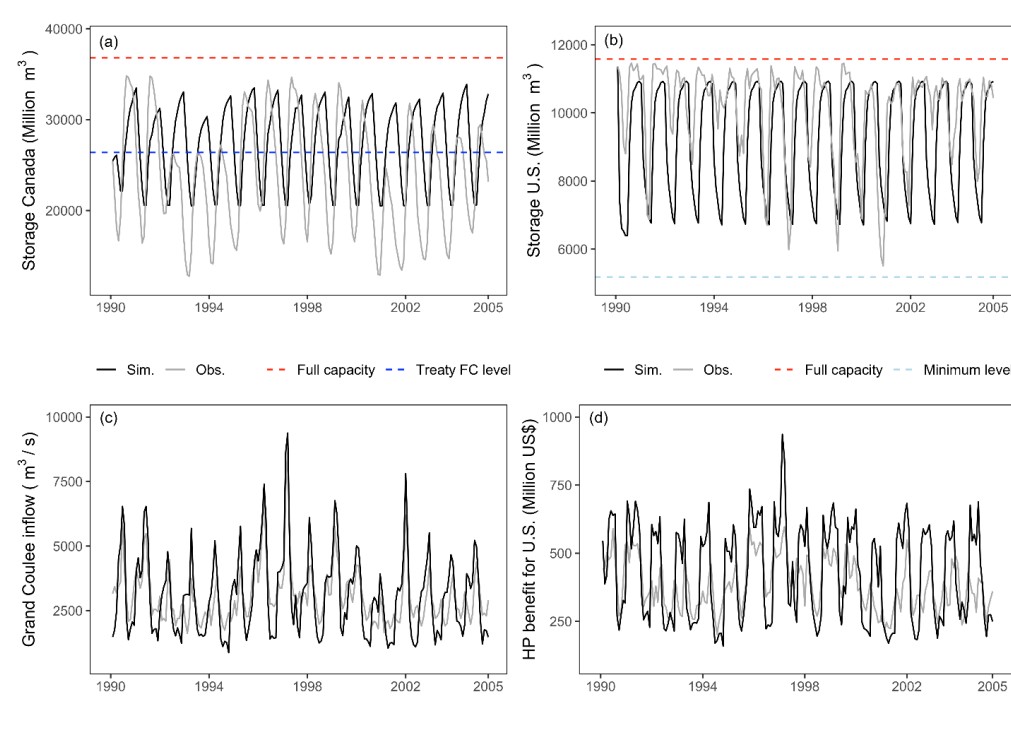


**Figure 5.** Calibration result from 1990-2005 showing, (a) Canadian storage, (b) U.S.

storage, (c) Grand Coulee inflow and (d) hydropower benefit for the U.S.


The model validation period was 12 years from 2006–2017 (Fig. 6a–d). Since the

warmup period during the calibration and validation simulation is only 3 months (i.e.,

when model stability is achieved), the selected calibration and validation periods are long

enough to yield robust results. Compared to calibration results, model validation

presented slightly better results in terms of performance metrics (Table 4). The simulated

behavior of the reservoir level in Canada and the U.S. during calibration and validation

are quite similar (Fig. 6a–b). In Canadian reservoirs, the model accurately simulates the

maximum peaks, but the simulated low reservoir level is higher than the observed (Fig.

5a and Fig. 6a). Meanwhile, for the U.S. reservoirs, the simulated lower reservoir level is

lower than observed (Fig. 5b and Fig. 6b). It is to be noted that the actual operating rules

for these dams are dynamic based on seasonal changes and weather forecasts. In practice,

they may change suddenly from the pre-determined plan given unforeseen circumstances.





Therefore, it is impossible to capture the exact behavior in a lumped model of this kind.
The validation result for Grand Coulee inflow (Fig. 6c) and hydropower benefit for the
U.S. (Fig. 6d) showed similar performance as the calibration period.

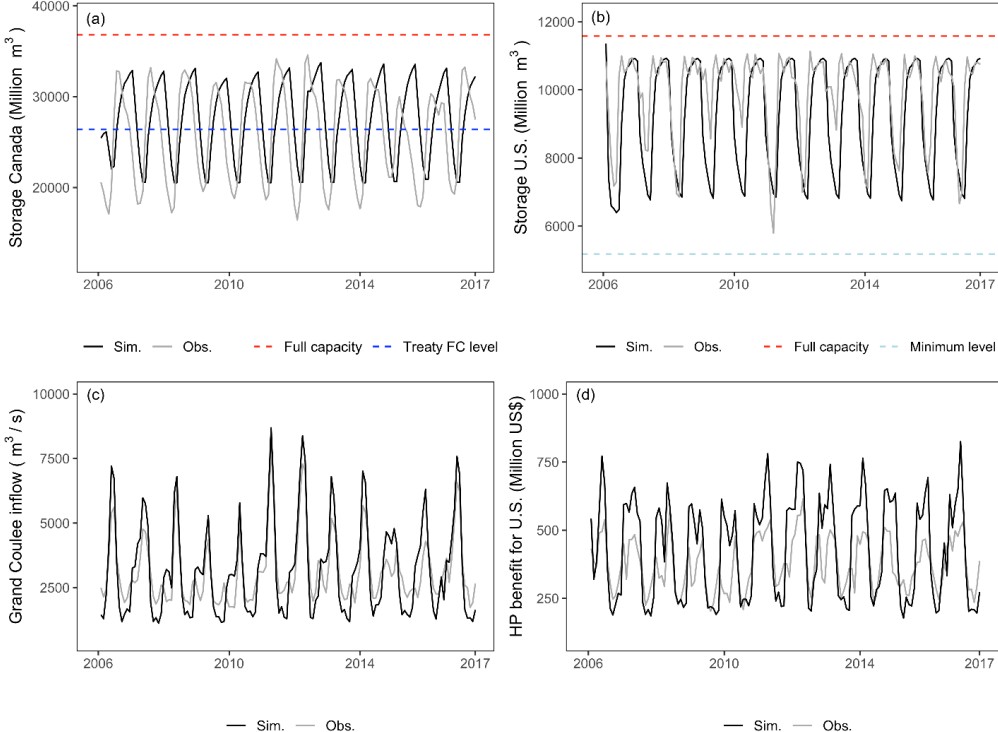


**Figure 6.** Validation result 2006 – 2017 showing, (a) Canadian storage, (b) U.S.
storage, (c) Grand Coulee inflow and (d) hydropower benefit for the U.S.

PBIAS for both calibration and validation showed that the result is close to
optimal, and Grand Coulee inflow showed the best fit with the PBIAS value that is closest
to 0. VE is only applied to the reservoir volumes and streamflow, as per the suitability of
the metric. VE values are greater than 0.72, suggesting a good fit. Similarly, agreement
index or rd values indicated better performance for all the comparisons except for
Canadian storage. The result of these metrics show that the model is able to replicate and
predict the desired behavior.

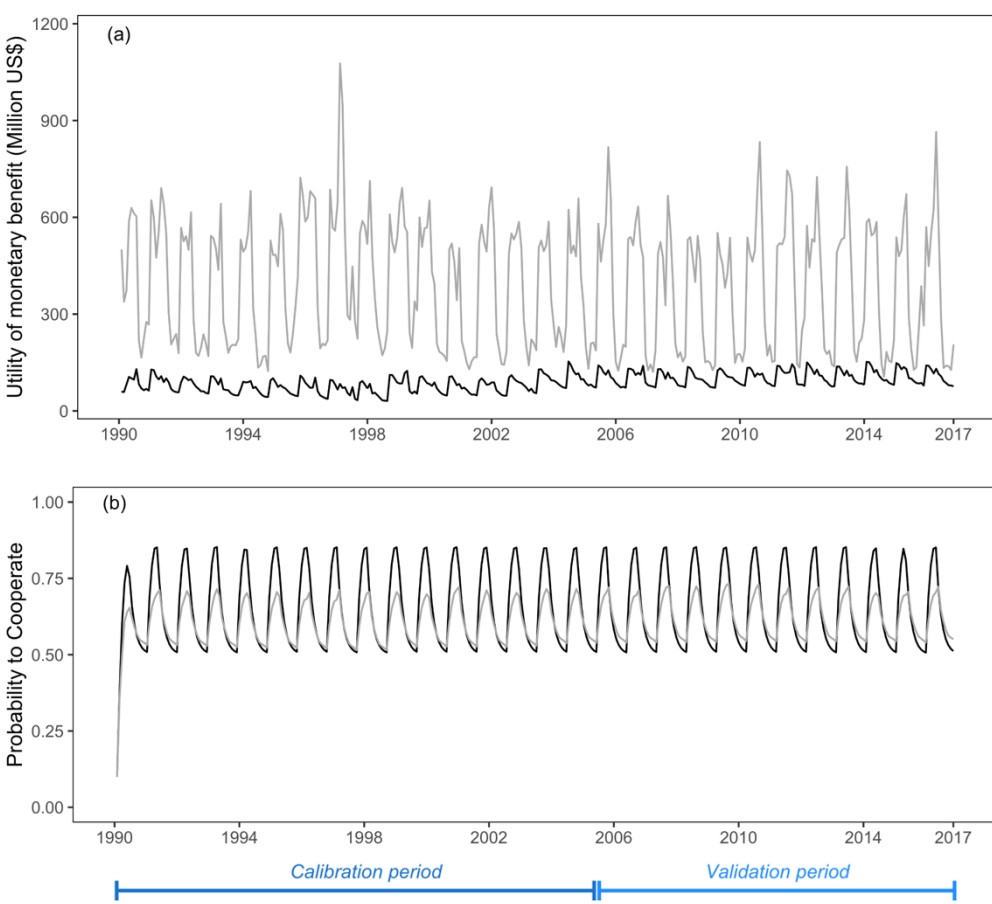

**Figure 7.** Change in, (a) the utility of monetary benefit and (b) probability to cooperation during calibration and validation period for Canada and the U.S. Note: The lower initial probability to cooperate during 1990 is only due to the warmup period of model simulations.

Figure 7a–b shows the utility of monetary benefit and dynamics of the probability to cooperate for the U.S. and Canada during the calibration and validation periods. This model simulation with calibrated parameters over 1990 to 2017 is also referred to as baseline in the next section. The share of benefits that the U.S. receives is higher than the benefit in Canada, relatively, despite the Canadian Entitlement (Fig. 7a). The minimum probabilities to cooperate for both countries converge at 0.5, while peak amplitude for cooperation dynamics is higher for Canada compared to the U.S (Fig. 7b).



### *4.2 Scenario analysis*


The scenario analysis results presented below is based on environmental and
institutional change, and social preferences. The scenario analysis covers the same time
period from 1990 to 2017, utilizing observed inflow, tributary streamflow, and storages,
and the same initial conditions as these simulations are not for projection, but rather to
gain a deeper understanding of dynamics in the socio-hydrological system.

### *4.2.1      Scenarios based on environmental and institutional change*


The four scenarios tested here are based on changes in environmental and
institutional conditions. The results are compared with the baseline scenario which
represents cooperation between both countries. In the quantile-quantile plot (Fig. 8a–f),
the baseline scenario is shown on the horizontal axis and four scenarios on the vertical
axis, where each point represent a time step. The scenario "$\chi$ *decreases*" significantly
reduces the probabilities to cooperate for both countries as the maximum $Cca$ reduced
from 0.85 to 0.7 and maximum $Cus$ reduced from 0.75 to 0.64.  The probability to
cooperate for Canada under the "$\chi$ *decreases*" scenario is identical to the "$\chi$ *decreases*
*and high fish spills*" scenario (Fig. 8a), thus blue and cyan points overlap. Reducing $\chi$
showed two distinct characteristics: the rise of $Cca$ and $Cus$ took almost 8 years of
simulation to converge and level off (which is not shown in the figure), although the
average value when the convergence occurred did not deviate much (thus values around
0.55 falls near the y = x line), the maximum probability to cooperate or $Cca$ and $Cus$
reduced significantly. Similar results were seen for the U.S. probability to cooperate (Fig.
8b). Lowering the $\chi$ resulted in lower $Cca$, and, therefore, Canada would be expected to
increase the level of storage in its dams to produce more hydropower as compared to
baseline (Fig. 8c). Lowering the $\chi$ impacted $Cus$ too, along with $Cca$, because, if Canada
increased its hydropower production, the U.S. would have to provide its own flood
control. Therefore, reservoir levels in the U.S. would decrease as compared to baseline
when $\chi$ decreases (Fig. 8d). Since Canada would produce its own hydropower in this
scenario, the monetary benefit increased slightly compared to baseline, and the result is
similar to the "$\chi$ *decreases and high fish spills*" scenario for Canada (Fig. 8e).

The "$\chi$ *increases*" scenario indicates better institutional capacity that favors
cooperation. Increasing $\chi$ increased the maximum probabilities to cooperate (i.e., $C_{ca}$ and


$C_{us}$) but the minimum remains the same (as lower quantile falls on the identity line or y
= x line) (Fig. 8a–b). While not shown in the figure, the time it took to converge is similar
to the baseline. With increasing $\chi$ Canada would provide flood control to the U.S. as
agreed upon in the CRT. Here, a slight increase in the capacity for flood control in
Canadian storage was observed in the model, as storage level decreased slightly below
the baseline (Fig. 8c) and the U.S. continues its existing operations to produce maximum
hydropower, hence the storage level in the U.S. remains the same as in the baseline (Fig.
8d). With increasing $\chi$, Canada's and the U.S.'s benefit continues to be the same as the
baseline (Fig. 8e). When $\chi$ increases or decreases the utility benefit that the U.S. receives
does not change significantly. This is due to the U.S. balancing the increased flood
damage control while hydropower production is compromised.

The "*High fish spills*" scenario refers to strict regulation to protect fish passage

along the Columbia River, which has negative implications for hydropower production.
Increasing fish spills in U.S. dams has no effect on the Canadian probability to cooperate
($C_{ca}$) as it does not affect Canadian dam operation (Fig. 8a). Increasing the fish spills
decreases peak $Cus$ slightly but the average remained similar to the baseline (Fig. 8b).
This also does not affect the storage level in the U.S. dams (Fig. 8d), but monetary benefit
for the U.S. decreases due regulation as water is diverted from the hydropower turbines
(Fig. 8f). It is to be noted that this loss of hydropower production affects the U.S. but has
no effect to Canadian benefit because the U.S. remains obligated to pay the Canadian
Entitlement even if hydropower production is lower. The combined scenario of *"$\chi$*
*decreases and high fish spills"* has similar results to the "*$\chi$ decreases*" scenario (Fig. 8a–
e), but reduction in monetary benefit is higher compared to the "*$\chi$ decreases*" and "*High*
*fish spills*" scenarios.





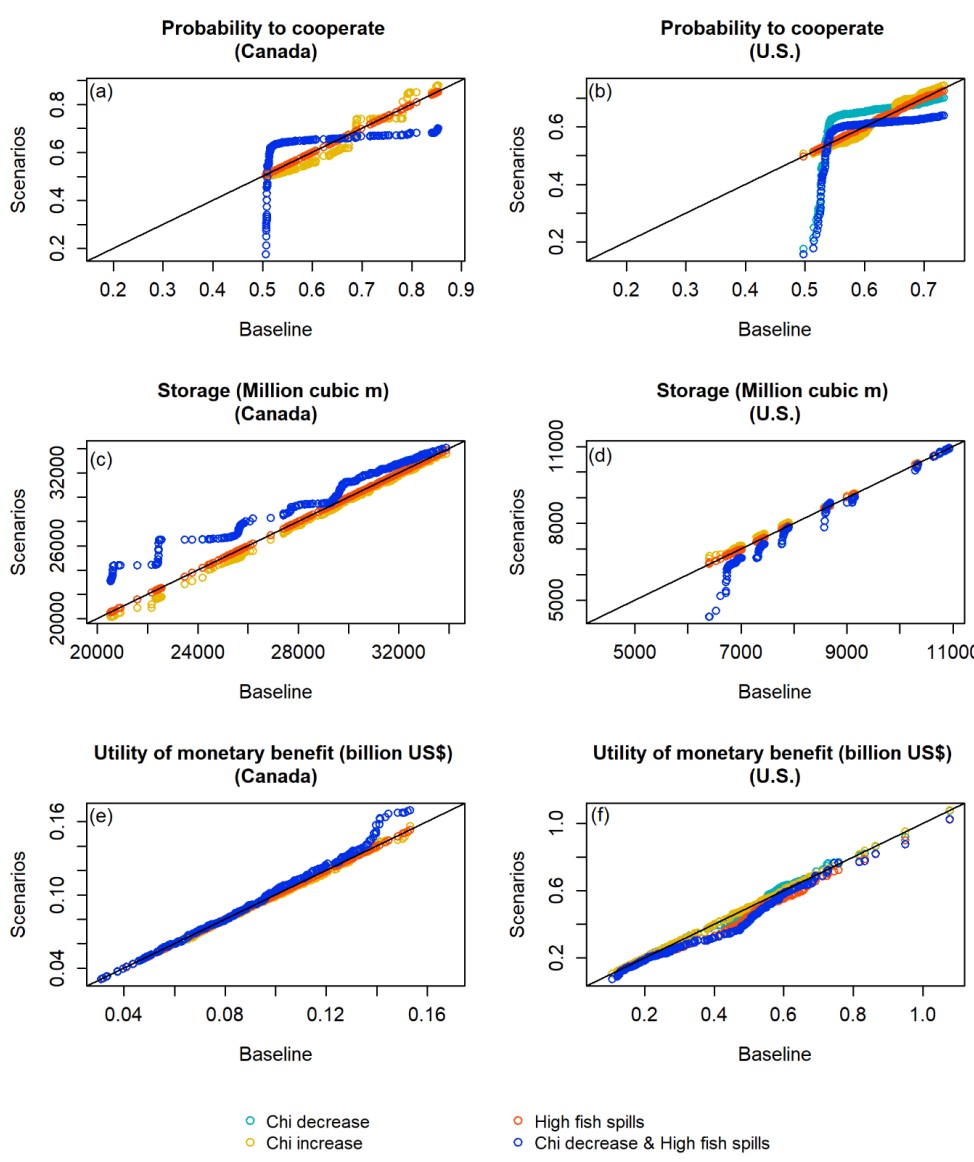

**Figure 8.** Quantile-Quantile plot of the baseline versus other scenarios ($\chi$ decrease, $\chi$ increase, high fish spills and combined $\chi$ decrease and high fish spills) comparing probabilities to cooperate, reservoir storage volumes and utility of monetary benefits

### 4.2.2    *Scenario analysis in terms of social preferences*

In addition to the scenarios above, four different scenarios of social preferences were tested and compared to each other. Figure 9 shows the differences between the





expected utility of cooperation and non-cooperation from each country according to
different scenarios.

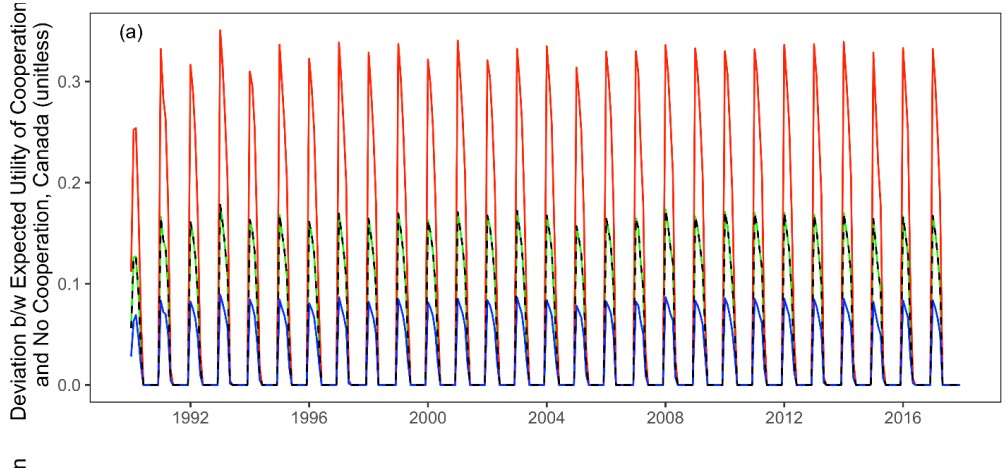

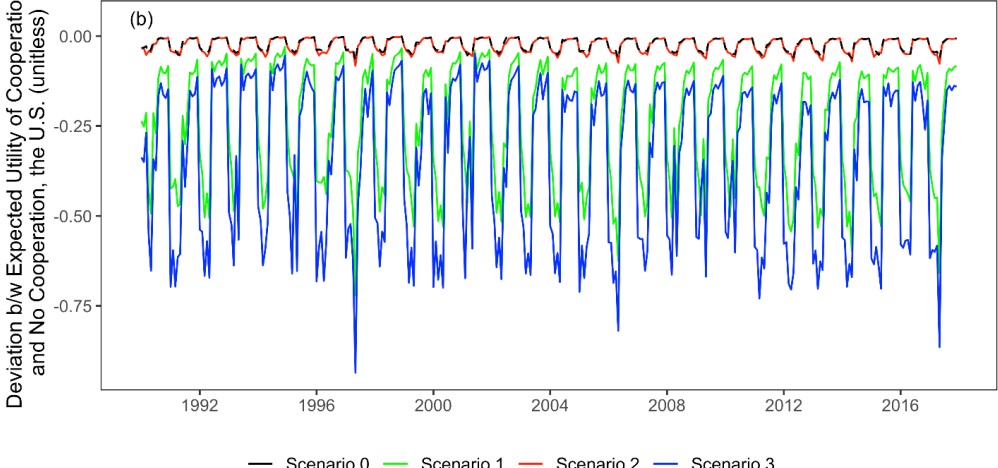


**Figure 9.** The differences between the expected utility of cooperation and no

cooperation from each country according to different scenarios for (a) Canada and (b)

the U.S.


Figure 10a–c, shows the changes in the probability to cooperation ($C_{ca}$ and $C_{us}$)

according to the different configurations of social preferences. As shown in Fig. 10a–c,
Canada's probability of cooperation is always higher than 0.5 in all scenarios because
Canada can get higher expected utility when it chooses to cooperate no matter which
behavioral types the two countries possess. This explains why the probability to cooperate



in Canada is always higher than the U.S. in Fig. 10a–c. Conversely, since the expected
utility of cooperation in the U.S. is always smaller than the expected utility of non-
cooperation in Fig. 9b, the probability of cooperation of the U.S. is always less than
Canada (Fig. 10a-c).

Comparing *"Scenario 0"* and *"Scenario 1"* from the standpoint of Canada, we
found that there was no difference in the outputs between *"Scenario 0"* and *"Scenario*
*1"* (Fig. 10a). This means that a decrease in the guilt coefficient of the U.S. does not affect
Canadian decision-making on whether to cooperate or not. However, in *"Scenario 2"*,
the gap between the expected utilities with cooperation and without cooperation widens
and Canada is more likely to continue cooperating when Canada feels more jealousy
(more sensitive to disadvantageous inequity) (Fig. 9a). From the standpoint of Canada, it
is always economically beneficial to cooperate with the U.S. because Canada can receive
the Entitlement from the U.S. under the CRT. In other words, the more unfair the
distribution of material benefits between Canada and the U.S., and the greater the jealousy
of Canada, the more Canada will be motivated to cooperate due to the Entitlement (Fig.
10b). In *"Scenario 3"*, the differences between the expected utility of cooperation and
non-cooperation decreases compared to *"Scenario 0"* if Canada does not care about the
counterpart's payoffs and focuses on its own payoffs (Fig. 9a). Cooperation will decline
as Canada is narrowly self-interested in the fair distribution of material payoffs (Fig. 10c).
In terms of cooperation, selfishness is worse than jealousy.

From the standpoint of U.S., there was no difference between "Scenario 0" and
"Scenario 2" in terms of outputs (Fig. 10b). This implies that a rise in Canada's jealousy
coefficient has no effect on the decision of U.S. whether to cooperate. Comparing
*"Scenario 0"* and *"Scenario 1"*, the difference between expected utilities with and
without cooperation is expanded, but the expected utilities of non-cooperation are larger
than those of cooperation (Fig. 9b). As a result, the U.S. is less inclined to cooperate in
the future when it feels less guilty (less sensitive to advantageous inequity) (Fig. 10a). In
other words, the more material benefits Canada receives and the less guilt the U.S. has,
the more driven the U.S. will be motivated to break the Treaty. Like *"Scenario 3"*, if the
U.S. does not care about the counterpart's payoffs and focuses on its own payoffs, the
relative magnitude of expected utility of cooperation will decrease. As the guilt of the
U.S. decreases, the U.S. becomes less concerned about a "fair deal" with Canada and





loses the motivation to continue cooperation. Therefore, the U.S. can maximize its profits
by halting cooperation (not paying the Canadian Entitlement) and operating unilaterally.

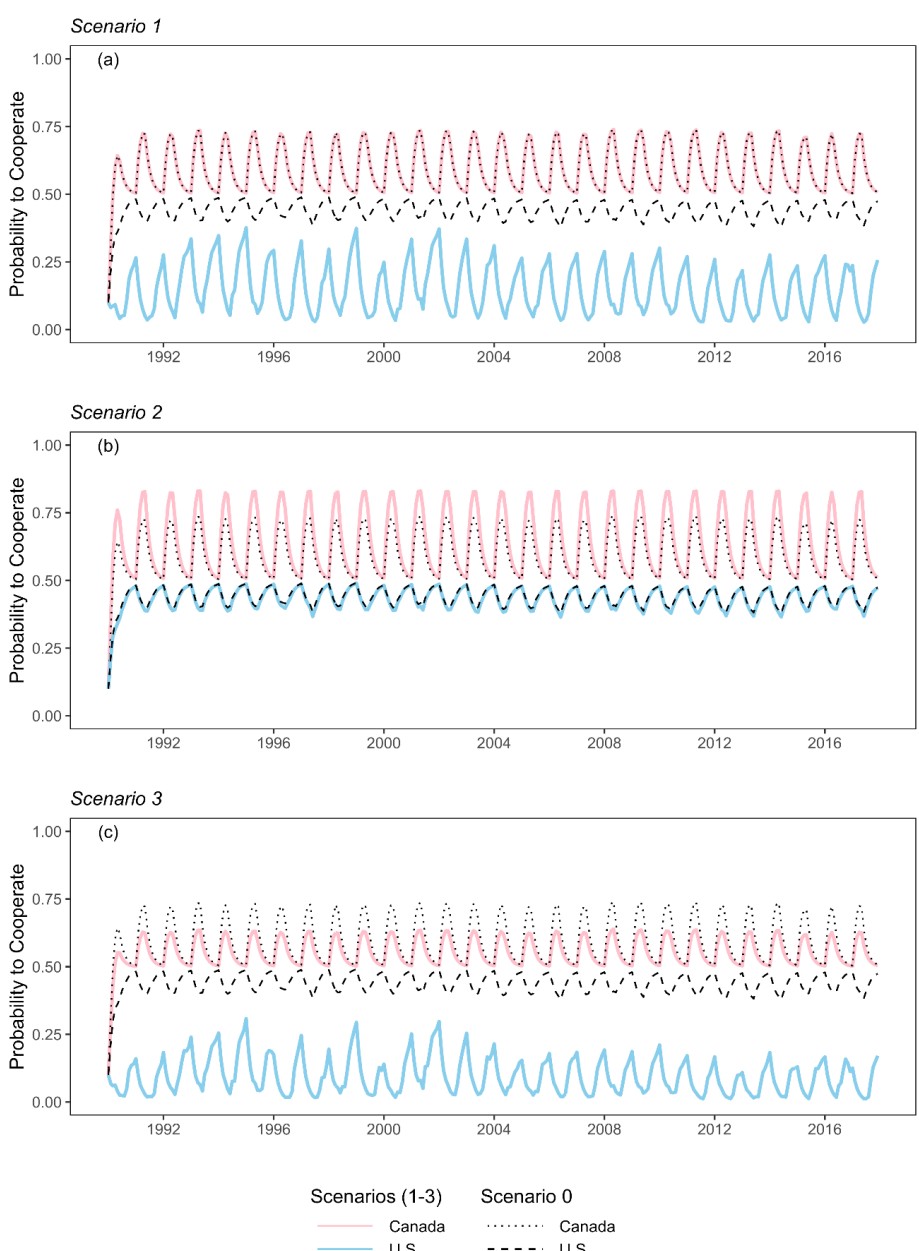


**Figure 10.** The probability to cooperate of each country according to different scenarios
(a) Scenario 1, (b) Scenario 2, and (c) Scenario 3

Since Canada gets the Entitlement due to the CRT, Canada is likely to continue
cooperating. If the U.S. preference for a fair distribution of benefits declines during future
CRT negotiations, such as in *"Scenario 1"* and *"Scenario 3"*, the U.S. is more likely to
break the treaty or change its stance on the Entitlement. That does not mean that the U.S.
has zero or negative benefit from the CRT. The U.S. has some benefits, but it would not
continue to cooperate because the benefits of not cooperating are greater than the benefits
of cooperating. As environmental concerns increase, the net benefit of the U.S. is
expected to decline further because of lower hydropower benefit, so the U.S. is less likely
to agree with continuation of the treaty until it is changed to create greater benefits for the
U.S. from cooperation.

**5     Discussion and conclusion**
The CRT is regarded as one of the most successful transboundary river
agreements. As the upstream and downstream actors, Canada and the U.S. have
asymmetric access to water resources, and different positions with regard to the risk of
floods and potential for hydropower production. Within the Columbia River basin,
Canada is less susceptible to flood risk relative to the U.S. and the U.S. has capacity for
higher hydropower production relative to Canada. The unique feature of the CRT is that
the two countries developed a plan to manage the river as a unified system and to share
the costs and benefits equitably (Bankes and Cosens, 2013; Shurts and Paisley, 2019).
This collective sharing of risks from flooding and benefit from hydropower as indicated
by Wolf (2007) and Zeitoun et al. (2013) makes the CRT successful among other
transboundary river treaties. This study examines the dynamics of cooperation, and how
it is affected by feedback between human and natural systems. It is important to
understand the underlying drivers of a successful cooperative regime and the factors that
influence each country's choice about whether to cooperate or not. The provisions of the
CRT expire in 2024, and negotiations for the next phase of the treaty are ongoing. There
have been many prominent discussions about what the future of the treaty should look
like, including issues related to hydropower generation versus fish, and how to account
for spills (Blumm and Deroy, 2019; Harman and Stewardson, 2005; Leonard et al., 2015;
Muckleston, 1990; Northwest Power and Conservation Council, 2019; United States
Government Accountability Office, 2018). Additionally, both countries perceive
imbalances in the benefits that are received from the CRT relative to what each deserves
or compared to what they perceive the other side's benefits to be (Holm, 2017; Stern,





2018). As discussed in Gain et al. (2021) and Gober and Wheater  (2014), the success in
treaties or institutions managing river basins depends not only on the control of hydrology
but in consideration of socio-political dynamics. This study shows that addressing
emerging social and environmental issues are critical to continued cooperation, providing
valuable insights for the current renegotiation process, as well as future treaty negotiations
on transboundary waterways similar to the Columbia River.

Natural and social systems evolve over time. Under unforeseen and uncertain

changes, the balance of these systems could shift. A subtle social change can be induced
by environmental and hydrological changes, which in turn lead to further unforeseen
changes in hydrologic or physical systems. For the Columbia River Basin sudden change
in cooperation and deviation from cooperation to conflict is not anticipated because both
countries that have similar economy and political power, and have shared values,
common interests and multi-layered economic ties. The socio-hydrological system
dynamics model developed for this study captures the dynamics of cooperation to reflect
external perturbations. Explicitly incorporating the probability to cooperate  $C_{CA}$ and $C_{US}$
(Eq. 5 and 6) into the model, enables exploration of the factors influencing cooperation.
This study further illustrates the utility of simplified lumped models in understanding
complex systems.

This socio-hydrological model presented here further allowed for the exploration

of scenarios under environmental and institutional changes, and social preferences, to
understand how robust the cooperation on this transboundary waterway is. These
scenarios represent current and plausible future socio-political and environmental
changes. We found that institutional capacity ($\chi$) plays an important role in long term
cooperation (Fig. 8a–b and Fig. S17, supplementary material (SI 3)). Stronger
environmental regulation for increased fish spills affects the benefit for the U.S. but not
as substantially as when $\chi$ (institutional capacity) decreases. Canada continues to receive
payment through the Canadian Entitlement, even when the U.S. is producing less
hydropower, something that is interesting to explore further for future negotiations of the
CRT. Different configurations of social preferences for the behavioral model of Canada
and U.S. was used to demonstrate how the probability to cooperate changes. The expected
utility of cooperation as compared to expected utility of non-cooperation is higher for
Canada and lower for the U.S. (Fig. 9). Thus, the probability to cooperate was simulated



to be higher for Canada. The results show that both the guilt coefficient of the U.S. and
the jealousy coefficient of Canada affect the level of cooperation. For future CRT
negotiations, the ideas considered in this study could help provide insight into the long-
term dynamics of cooperation and the impacts of benefit sharing. For other transboundary
rivers (e.g., along Nepal and India, Bangladesh and India, or India and Pakistan (Ho,
2016; Mirumachi, 2013; Saklani et al., 2020; Thomas, 2017; Uprety and Salman, 2011)),
the jealousy and guilty coefficient between actors and their social preferences will not be
the same as in Columbia River Basin. Similarly, the tipping points for the balance of
cooperation arising from environmental and social change could be different and this
warrants future research in other transboundary river basins.

This socio-hydrological system dynamics model can be further improved by

considering additional variables related to climate change, land use change and water use
regime changes. The key limitation of this study is the explicit consideration of water use
for hydropower production and flood control only. The study does not consider future
projections of these variables, which would be a possible direction for future research.
Another limitation is the method of estimation of flood damages. We estimated the
economic benefits involving flood damage prevention, which does not include the
monetary benefit of flood control in Canada due to treaty dams because little information
is available in the scientific literature and official reports, and existing resources indicate
significantly less flood damage in Canada relative to the U.S. (BC Ministry of Energy
and Mines, 2013; Northwest Power and Conservsation Council., n.d.). However, future
studies should investigate the magnitude of this benefit since there are certainly flood
risks averted by Canadian storage.

As mentioned previously, the results of this study can help inform the

renegotiation of the CRT in two ways: (1) the methods of modeling the hydrological and
social systems in tandem, and using behavioral economics, could be used to help
formulate policies or management priorities and (2) understanding of the connection
between the share of benefits received by each side to cooperation can support negotiation
discussions to find solutions that would satisfy both sides. More generally, the model
demonstrates that understanding the motivations of each country in terms of guilt and
jealousy might provide insight into the factors driving each country and the thresholds


that might influence their decision about whether to cooperate. We also find that it is of great importance to maintain institutional strength in support of cooperation.

Unlike the U.S. and Canada where a non-cooperative regime or resort to direct conflict is unanticipated even if the benefits are perceived to be severely imbalanced, there are many other river basins where different environmental challenges are evolving (UNEP, 2016) and political tensions are high. Globally, conflicts do arise between countries that share a water source, with root causes that extend far beyond the water system (Sadoff and Grey, 2002). However, transboundary rivers support the livelihoods of millions of people, preserve ecosystems, and provide a vital resource that needs to be managed sustainably. Using the methodologies presented in this study and the insights gained could be applied to other river basins around the world to help us understand what behaviors and benefits are driving choices about cooperation.

**Author contribution**

AS, FS, SP and CC planned this work as participants of "Socio-Hydrology Summer Institute on Transboundary Rivers"; AS focused on model development and analysis; FS focused on data collection and data analysis; SP focused on behavior economics; CC focused on review and synthesizing Columbia River treaty; AS, FS, SP and CC conceptualized the system dynamics framework; FS and AS formulated stock and flow equations; SP formulated cooperation dynamics equations; AS and SP formulated hydropower and flood control benefit equations; CC conducted assessment of past and current issues affecting treaty renegotiation; AS wrote the model script, performed model testing, scenario analysis and data visualization; SP performed social preference scenario analysis and assessment;  AS, FS, SP and CC wrote the manuscript draft; AS revised the manuscript; MG, DY, and EM provided guidance and funding, and reviewed and edited the manuscript.

**Acknowledgement**

We acknowledge "Summer Institute on Socio-hydrology and Transboundary Rivers" held in Yunnan University, China in 2019, and Jing Wei for support and feedback.  We also acknowledge our professors - Giuliano Di Baldassarre, Günter Blöschl, Megan Konar, Amin Elshorbagy, Fuqiang Tian, and Murugesu Sivapalan for their feedback we received during and after the institute. A.S. was supported by M.G.'s startup funds from





Arizona State University. M.G. was supported by the National Science Foundation
grant: Cross-Scale Interactions & the Design of Adaptive Reservoir Operations [CMMI-
1913920]. SP and DY were supported by NSF CMMI 1913665 and a Purdue Research
Foundation (PRF) Grant.

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
