# Peer review of "Socio-hydrological modeling of the tradeoff between flood 1"

_Hydrology and Earth System Sciences, 2021_

## Referee Comment (RC2)

This manuscript develops a socio-hydrological model to simulate the cooperation dynamics of flood control and hydropower in Columbia River Basin on basis of Columbia River Treaty (CRT) signed between the United States and Canada. Overall, it's an interesting study within the scope of socio-hydrology and transboundary rivers, and the proposed model has potential application value in other basins. However, I have some concerns and suggestions, which needs to be addressed. Below are detailed comments:

**Major concerns**

1. It's unjustified that the authors linearly aggregated the reservoirs for flood control and hydropower production. Flood control and hydropower production not only depend on reservoirs operation rules, but also related to the hydrological connections between reservoirs. The aggregated reservoirs may be applicable for the total storage, but will be bound to bring risks on flood control and hydropower production.

2. The flood damage is typically estimated based on the peak daily water flow in a year. However, I notice the proposed model in study conducted with a monthly time step, which indicates that the peak daily water flow have been smoothed. The flood damage will be thereby remarkably underestimated, significantly challenging current results.

**Minor concerns**

1. How to distinguish the positive and negative feedbacks between variables in Figure 2?

2. I am puzzled about equation (3) and (4):

(1) The simplified reservoir operation rule indicated by equation (3) and (4) is used to determine the outflow, which is considered as vital factor in the model. It's suggested to cite corresponding references and add justification description for these equations.

(2) It's worth noting that $n_{CA}$ is an important parameter for outflow of Canada. What's the explicit connotation of $n_{CA}$ and how to determine it?

(3) The outflow is dominated by storage thresholds (i.e., $S_{CAthreshold}$ and $S_{USthreshold}$). The storage threshold is always between the target flood control storage ($S_{FCthreshold}$) and target hydropower storage ($S_{HPthreshold}$) as shown in Figure 3, as storage threshold is estimated by linearly aggregating $S_{FCthreshold}$ and $S_{HPthreshold}$ in equations (5) and (6), which is prone to simultaneously increase flood damage and decrease hydropower production. Please give more justification description.

3. Please check whether the second '$C_{CA}$' is a typo in equation (6).

4. The motivation of applying logit dynamics functions to simulate the cooperation probability variables $C_{CA}$ and $C_{CA}$ should be detailed in line 378.

5. It's unjustified to determine the hydropower without considering water head in equation (20), despite that the simulated series can fit the observed series well. Moreover, the threshold water flow is directly selected as 400 m$^3$/s, which needs more description.

6. It's suggested to add another section in Methodology to describe the feedback loops on basis of the dynamic equations in Section 3.2.

7. In line 677, how to determine whether the stability is achieved?

8. In Figure 7(b), the trajectories of probability to cooperate perform notable periodicity, which needs to be well accounted.

---

## Author Response (AR1)

The manuscript titled "Socio-hydrological modeling of the tradeoff between flood control and hydropower provided by the Columbia River Treaty" by Shrestha et al. understand the cooperation dynamics in Columbia River Basin through investigating how and what factors drives the two countries into a successful cooperative regime in the past, and what would the balance shift in face of the social, institutional and environmental changes. The paper is generally well written and structured. The concept of the paper is interesting, and crucial one for understanding the underlying mechanism of a successful cooperation dynamic and transboundary co-evolutionary dynamics in general. On top of that, this study provides valuable insights and reference for the negotiations of the treaty within and beyond Columbia River. I recommend this paper being accepted with some minor revisions.

*General Comments:*

- The manuscript can generally be improved with a more solid literature review in the introduction. More specifically the authors are encouraged to review on the existing studies in understanding transboundary rivers management from different disciplines, and through the lens of conflict and cooperation dynamics. The selection of variables that influence on the choice of cooperation, i.e. institutional capacity, social and behavioral preferences could be articulated.

*Authors' response*: Thank you for your kind evaluation and helpful comment. We will articulate the cooperation and conflict dynamics in other transboundary river basins based on the existing literatures from the perspective of institutional capacity and social preferences.

*Revision (#1.1):* We have done additional literature review and summarized how several studies have addressed transboundary water management. The revised text as included in the "Introduction" of the revised manuscript (lines 88 - 115) is as below:

...Transboundary water management compounds the challenges of managing water between competing users because the river is managed between different jurisdictions and under different policy structures (Bernauer and Böhmelt, 2020). Transboundary water management has been studied through different disciplines. Kliot et al. (2001) reviewed the institutional evolution of the water management in twelve transboundary river basins identify legal principles that organize transboundary water management and discuss their characteristics and shortcomings. The authors discuss that the key challenges in transboundary water management arise from water scarcity, maldistribution, over-utilization and misuse of shared resource. Odom and Wolf (2011) examined the 1994 Israel-Jordan Treaty of Peace where climate extremes and drought created conflicts on water sharing and hydropower agreements, but the modified institutional arrangements mitigated conflicts and vulnerabilities in transboundary water management under climate change. Madani et al. (2014) applied bankruptcy resolution methods to the challenge of water allocation in transboundary river basins. This quantitative approach is rooted in the economic literature and offers insight into efficient and stable allocation schemes. Pohl et al. (2017) posit that transboundary waters create economic, social and environmental interdependencies that can be leveraged to either promote cooperation or intensify conflict. They highlight that this creates the potential for broader peace dividends when negotiating

transboundary water management and present strategies for diplomats to engage constructively. Islam and Susskind (2018) presented the Water Diplomacy Framework which draws on the concepts of complexity science (e.g., interconnectedness, uncertainty and feedbacks), and negotiation theory (e.g., stakeholder identification, engagement at multiple levels, and value creation for benefit sharing), to understand and resolve transboundary water issues and cooperative decision making. Koebele (2021) takes a policy process approach to understand collaborative governance in transboundary water management of Colorado River between the U.S. and Mexico, where overallocation of water led to environmental problems and water scarcity downstream. The author applies the Multiple Streams Framework, used to explain decision making in a range of policy contexts, to examine the case of transnational policymaking in the Colorado River Delta. External factors such as climate change affect the sustainable transboundary water management.

*Detailed comments:*

> - Line 49: "actors' decisions are guided by their or social preferences", delete "or";

*Authors' response***:** Thank you for pointing this out. We will improve this in our revised manuscript.

***Revision (#1.2):*** The correction was made as suggested.

> - In the introduction line 50-52, the authors stated that "actors exhibit social preferences if the actor not only cares about their own material benefit but also cares about the material benefits of other actors", this is not clear, please re-structure this sentence.

*Authors' response***:** Thank you for this comment. The sentence is conveying that, as suggested by Fehr and Fischbacher (2002), and Kertzer and Rathbun (2015) the decision makers have social preferences and that their decision is motivated by social preferences, which is the behavioral characteristics that such actors care about gain (here, material payoff) not just for themselves but also for others. We will certainly revise this.

***Revision (#1.3):*** We have revised the text in lines 52 - 55.

... preferences). Fehr and Fischbacher (2002), and Kertzer and Rathbun (2015) suggest that decision makers have social preferences that motivate their decisions, which means that such actors care about gain (here, material payoff) not just for themselves but also for others. The perceived fairness of ...

> - Line 64: update the number of global transboundary river basin with 310 rivers, see McCracken & Wolf 2019 for the most updated info on this:
>   "Updating the Register of International River Basins of the world" by McCracken & Wolf 2019, https://doi.org/10.1080/07900627.2019.1572497

*Authors' response***:** We appreciate updating us about this recent information. We will revise this.

***Revision (#1.4):*** We have revised the text in lines 82 - 83.

Globally, 310 international transboundary river basins cover almost 47.1% of the Earth's land surface, which includes 52% of the global population and are the source of 60% of freshwater supplies (McCracken and Wolf, 2019; UN-Water, 2015; United Nations, n.d.). Transboundary water...

> - Line 70, what is "social comparison"?

***Authors' response***: Social comparison refers to the social behavior that actor compares their position, benefit, or risks with other actors. For example, according to some previous research in behavioral economics, it was empirically revealed in a field experiment that people tend to be more cooperative if they know many others are contributing (Frey and Meier, 2004). We will elaborate this by explaining it alongside the social preferences.

***Revision (#1.5):*** We have added new text and rearranged the paragraph through lines 68 - 80.

...and downstream riparian states; those political dynamics are shaped by social comparison in which actors compare their position, benefit, or risks with other actors (Gain et al., 2021; Gober and Wheater, 2014). Research in behavioral economics by Frey and Meier (2004) has shown that actors tends to be cooperative if they know many others are contributing too, which could be key to successful management in transboundary river basins. Transboundary rivers are managed by multiple heterogeneous stakeholders with different sovereignty, governance structures and economic conditions; while diverse, basin populations may be interdependent not just hydrologically but also economically and socially (FAO, n.d.; Rawlins, 2019). Social factors that can explain cooperation and conflict dynamics include asymmetric access to water resources due to upstream-downstream locations, and varying levels of dependence on different uses of the river (Warner and Zawahri, 2012).

> - Paragraph 89 – 100 introduced the challenges of cooperation in transboundary river basins through listing the possible impacting factors, i.e. political/economic power, geographic locations, followed by the four types of benefits, which were a bit of a sudden jump, please consider re-structure this paragraph.

***Authors' response***: Yes, we agree. We will revise these two arguments with better transition.

***Revision (#1.6):*** We rearranged the paragraph and revised the text through lines 125 – 133 as below:

Understanding the history of such transboundary river basins where conflicts prevailed more than cooperation showed that there are inequitable distribution of benefits and risks among actors. In the absence of cooperation, the benefits and risks are usually distributed in advantage to actors with higher political and economic power or following geographic advantages (Dombrowsky, 2009). Prevalence of such imbalance in benefits and risks could further diminish the likelihood of successfully negotiating any agreement to cooperatively manage water

resources (Espey and Towfique, 2004; Song and Whittington, 2004). In case of cooperative transboundary river management, actors mutually achieve several benefits, including: ...

> • Line 121- 135, descriptions on social preferences, there are four types of social preferences stated, what are the differences between the social preferences and social motives? There are also four types of social motives: individualism, competition, cooperation and altruism, how is the social preference differentiate with the social motives and why social preferences is selected here?

*Authors' response***:** Thank you for this important question. Considering the equal benefits and risks sharing provision between two actors in CRT, each actor has their individual decision roles which determine the benefit or risk they receive. The successful continuation of CRT is indeed the result of mutual decision making and agreement, rather than competition. And as the agreement was founded on equal benefit sharing, the two actors and their decisions are best described by inequality aversion.

*Additional response:* We realized that we misinterpreted your comment above.

*Revision (#1.7):* We revised our manuscript by adding a new paragraph in lines 157 – 184 as below:

The fairness consideration behind the CRT is consistent with the now well-established behavioral insight that most human actors are *not* selfish rational actors that seek to maximize short-term material benefits with complete information (Henrich et al., 2005). Rather, there is an overwhelming empirical evidence that humans are learning and norm adopting actors whose decisions are sensitive to contextual conditions, including that of how material benefits are relatively distributed between oneself and others (Fehr and Schmidt, 1999; Gintis et al., 2003). Among several social science theories that have emerged to explain this empirical regularity about human behavior (note that, as explained by Sanderson et al. (2017) the social sciences are characterized by theoretical pluralism and that there is no single universal theory about human behavior), perhaps the most rigorous theory is that of *social preference* which is also referred to as *prosocial preference* or *other-regarding preference* (Fehr and Fischbacher, 2002; Kertzer and Rathbun, 2015). This theory assumes that humans not only care about their own material benefits but also about the material benefits received by others, and that this intrinsic nature is consistent with why many people (but not all) exercise social norms such as inequality aversion and conditional cooperation. In line with this theory, the utility of individual and organizational actors can be formalized and categorized into four general types of social preferences: preference for having the benefits among all actors to be equal (inequality aversion), preference for maximizing group- or societal-level benefits (social welfare consideration), preference for rational self-interest maximization (homo economicus), and preference for having their own benefits to be higher than those of others (competitiveness) (Charness and Rabin, 2002). Among these four types, particularly relevant to transboundary river management is that human actors have a strong social preference for inequality aversion at both individual and organizational level, and that this preference is often a key to why cooperation emerges and is sustained among unrelated parties (Choshen-Hillel and Yaniv, 2011; Kertzer and Rathbun, 2015). Thus, the

decisions of organizational actors and their reciprocal interactions over time in the context of the CRT can be described and plausibly explained by inequality aversion.

> • Line 150, this research builds upon the work of Lu et al. (2021), could the author explicitly explain the novelty developed for the model used in this paper, what are the advancement?

*Authors' response*: The application of socio-hydrological model in the transboundary river basin to study the dynamics of cooperation between actors as an evolutionary process is relatively new in science of socio-hydrology, which is discussed by Lu et al. (2021) too, and we use the similar concept in the Columbia River Basin to study the dynamics in cooperation as the function of reservoir operation, equitable benefit sharing and feedback of this benefit sharing. Particularly in our study we used the concept of behavioral economics with social preferences between actors and convert overall benefits of water resource management to the utility of cooperation or no-cooperation. This allow us to quantify the cooperation for each actor as an individual decision maker. It is also to be noted that the power dynamics between actors is very different in Columbia River Basin than in Lancang-Mekong River Basin. We also simplified the structure of model such that these individual actors' cooperation directly affects lumped reservoir operations using continuous input of the streamflow (i.e., inflow) as the independent variable and other key variables such as outflow and benefits as the response variables. This approach of integrating concept of behavioral science such as social preferences is suitable particularly (and extendable) to cases when the reciprocity between actors is the main driver for cooperation, where system operates to share benefits equitably while ensuring the resources are sustainable.

*Revision (#1.8):* We revised the text through lines 205 – 208 as below:

...adhering to the agreement. Extending the work by Lu et al. (2021), we apply behavioral economics to incorporate the role of social preferences between actors to quantify the probability of cooperation for each actor. Furthermore, the power dynamics between actors is very different in Columbia River Basin than in Lancang-Mekong River Basin. The objective of this study...

We also added following text in "Discussion and conclusions" in lines 1025 – 1029 as below:

...warrants future research in other transboundary river basins. Our approach of integrating concept of behavioral science such as social preferences is suitable particularly (and extendable) to cases when reciprocity between actors is the main driver for cooperation, and where system operates to share benefits equitably while ensuring the resources are sustainable.

> • Figure 2, some variables illustrated in the figure are not explained, i.e. "utility for cooperation", "Utility for no cooperation", etc., also, the feedback loop illustrated could be improved by differentiating variables by different types, i.e. economic variable, hydrological variables, social variables, etc., to reflect the infrastructural, hydrological, economic, social, and environmental aspects being considered in this model.

*Authors' response*: Thank you for this suggestion. We will revise figure 2.

***Revision (#1.9):*** We have revised the figure 2 (in line 318) as below:

[revised manuscript text omitted]

This manuscript develops a socio-hydrological model to simulate the cooperation dynamics of flood control and hydropower in Columbia River Basin on basis of Columbia River Treaty (CRT) signed between the United States and Canada. Overall, it's an interesting study within the scope of socio-hydrology and transboundary rivers, and the proposed model has potential application value in other basins. However, I have some concerns and suggestions, which needs to be addressed. Below are detailed comments:

**Major concerns**

> 1. It's unjustified that the authors linearly aggregated the reservoirs for flood control and hydropower production. Flood control and hydropower production not only depend on reservoirs operation rules, but also related to the hydrological connections between reservoirs. The aggregated reservoirs may be applicable for the total storage, but will be bound to bring risks on flood control and hydropower production.

*Authors' response***:** Thank you for this critical comment. We understand that the risk of simplifying the physical system is that there could be model and output uncertainty. For this study lumping the storages for system dynamics model is appropriate for the reasons as discussed below:

1. The actual reservoir operations for the treaty dams is different from their designed operation plan and depends on the number of variables including future projection of streamflow. Modeling the exact reservoir operations for all the treaty dams is outside our scope of the research objective and is not necessary to investigate our research questions.
2. All the treaty dams work in coordination to achieve either of the hydropower benefits (by U.S. dams) or flood control (by Canadian dams). This coordinated operation supports that lumping the reservoirs is a valid approach.
3. The system dynamics model is not for prediction but inference of the observed data to understand the dynamics of cooperation along with the socio-hydrological system behavior. For this purpose, we have used the observed streamflow, particularly inflow hydrograph, to simulate outflow hydrograph and quantify the socio-economic variables (i.e., benefits) as a function of outflow.
4. In lumping the system, we have considered external input variables such as tributaries and added to the outflow from Canadian reservoir, or inflow to the U.S. reservoir. Using the calibration process, we have also ensured that the hydropower benefits is well captured.
5. The historical database for the flood control benefits or flood damages over the regular time interval is not available as per our knowledge and search. Thus, the most reliable approach we have adopted as discussed in "3.2.3 Economic benefit equations", using the data from the past study by Sopinka and Pitt (2014) to infer the flood control benefit is the appropriate option for this study. Moreover, the independent variable here is also the outflow hydrograph.

*Revision (#2.1):* We have added text in lines (331 - 336) as below:

...Grand Coulee dam is the only multifunctional dam with useable storage for flood control. We used the lumped reservoir approach to simplify the system process required to investigate our research questions. The lumped approach is particularly appropriate because all the treaty dams work in coordination to achieve either of the hydropower benefits (by U.S. dams) or flood control (by Canadian dams). In lumping the system, we have considered external input variables such as tributaries and added to the outflow from Canadian reservoir, or inflow to the U.S. reservoir. These dams along the Columbia River...

> 2. The flood damage is typically estimated based on the peak daily water flow in a year. However, I notice the proposed model in study conducted with a monthly time step, which indicates that the peak daily water flow have been smoothed. The flood damage will be thereby remarkably underestimated, significantly challenging current results.

*Authors' response***:** Thank you for this comment. We have selected monthly temporal resolution because some processes like increasing environmental (i.e., fish spill) flow is relevant in monthly time scale, as well as data such as hydropower production is available at the monthly time scale. The possible error in flood damage from using monthly streamflow is an important and valid point. We are exploring different approaches to address this. We will certainly check for this error and will address this in our revision.

*Additional response:* Again, thank you for this comment. We revised our model experiments by collecting the daily data from (Environment Canada, n.d.) including streamflow (inflow and outflow) and forebay level from various stations. To extract Mica Dam's inflows, stations IDs: 08NC004, 08NB014, 08NB005, 08NB019 were used. Mica Dam's water level was extracted from the station ID: 08ND017. For Duncan Dam's inflow and outflow, station IDs: 08NH119 and 08NH118 were used, along with the station ID: 08NH127 for Duncan Dam's water level. Stations IDs: 08ND013, 08NE006, 08NE077 and 08NE126 were used to extract Arrow Dam's inflow and outflow, and station ID: 08NE102 to extract Arrow Dam's water level. Station ID: 08NE058 was also utilized to extract streamflow at the international boundary between the Canada and the U.S.

The reservoir capacity table (i.e., storage vs. elevation curve) (USACE, 2013) was used to estimate daily storage volume for Mica, Duncan and Arrow Dams.

Similarly, daily U.S. treaty dams' inflow, outflow, forebay water level and hydroelectricity generated were extracted for Grand Coulee and other Dams using Water Control Data (USACE, n.d.). In addition, the streamflow at The Dalles, OR was extracted from USGS station ID: 14105700 (USGS, n.d.).

All the daily data were collected from the year 1990 to 2017. To ensure the data quality of the collected daily data, it was verified with the monthly average data collected and used earlier. The data gaps in the daily time series are minimal across all stations. Where required, any missing daily data was filled by using the long term daily average streamflow within the same station. Thus, the system dynamics model was simulated in the daily time steps.

***Revision (#2.2):*** Based on the use of daily data, a new relationship between streamflow at The Dalles and the Grand Coulee outflow + Snake outflow was established, which is given by the equation:

$$Q_{Dalles} = 1.3329 * (Q_{Grand\ Coulee} + Q_{Snake\ River}) - 122.91 \qquad (18)$$

The estimated relation was also revised in supplemental material Figure S13.

Similarly, a new relationship between electricity generation (MWh), and daily outflow + forebay level was established which is given in the equation:

$$HP_{Grand\ Coulee} = 0.042 * (Q_{out} * h) + 9802.7 \qquad (19)$$

The estimated relation was also revised in supplemental material Figure S14.

Due to the data gaps in daily electricity generated by 5 other U.S. dams (namely Chief Joseph Dam, McNary Dam, John Day Dam, The Dalles, and Bonneville Dam) existing equation was modified as below to estimate daily electricity generated.

$$HP_{5\ dams} = \begin{cases} 40.3 * (W_{fish} * Q_{out})\ for\ W_{fish} * Q_{out} \leq 4000 m^3 s^{-1} \\ 27.8 * (W_{fish} * Q_{out})\ for\ W_{fish} * Q_{out} > 4000 m^3 s^{-1} \end{cases} \qquad (20)$$

Thus, all the result figures 5 – 10 and table 4, that is based on the daily model simulation, are updated in the revised manuscript. The key results are quite similar to the earlier results.

Please refer to the revised manuscript for new figures 5 – 10 and table 4.

**Minor concerns**

1. How to distinguish the positive and negative feedbacks between variables in Figure 2?

*Authors' response***:** The causal loop diagram is revised and will be updated in the revised manuscript to include the loop polarity.

***Revision (#2.3):*** We have revised the figure 2 (in line 317) as below.

[Figure]

**Figure 2.** The causal loop diagram presents the hydrological and cooperation feedbacks between the Canada and the U.S. Different colors shows the hydrological, environmental, economic and social variables.

The following text was also revised to elaborate the causal loop diagram through lines 314 - 369:

The modeling framework is illustrated with a causal loop (CL) diagram in Fig. 2. The CL diagram illustrates all the key hydrological, environmental, economic and social variables, relationships, direction of those relationships and feedback.

The storage capacity of Canada (upstream) and the U.S. (downstream) are two important state (hydrological) variables which represent the aggregated storage of the treaty dams (Fig. 2),

the operation of which is determined by the storage thresholds. The increase in a storage threshold results in an increase in the storage level. Three Canadian dams namely Mica, Duncan and Keenleyside are lumped into a single storage as all three dams are multifunctional for flood control and hydropower production. In the U.S., the Grand Coulee dam is the only multifunctional dam with useable storage for flood control. We used the lumped reservoir approach to simplify the system process required to investigate our research questions. The lumped approach is particularly appropriate because all the treaty dams work in coordination to achieve either of the hydropower benefits (by U.S. dams) or flood control (by Canadian dams). In lumping the system, we have considered external input variables such as tributaries and added to the outflow from Canadian reservoir, or inflow to the U.S. reservoir. These dams along the Columbia River either have significant flood control capacity or significant hydropower production capacity (Table 1). Thus, the simplified reservoir operation described below in Sect. 3.2.1 was implemented in the lumped storages on each side of the border, which represent collective operation of all the treaty dams within each country. Other hydrological variables in the model (i.e., flows in the CL diagram) are inflow into Canadian storage, outflow from Canadian storage plus intermediate tributaries, inflow into the U.S. storage, and outflow from the U.S. storage. The higher the outflow from the dams, the lower the flood control as flood damages increase. A portion of the reservoir outflow passes through hydroelectric turbines, thus more outflow yields higher hydropower benefit. However, the need for flood control is intermittent depending on the seasonal high flows. Thus, Canada does not reduce the storage level throughout the year, but just before the incoming higher flows. Reservoir levels in the U.S. (under CRT) are kept as high as feasible to maximize hydropower generation. Each country's reservoir outflow is used to calculate flood control and hydropower production (Fig. 2, economic variables), which is converted into monetary units as shown in the CL diagram. Fish spill is included as an environmental variable as the reduced salmon migration causes depletion of the salmon population in Columbia River. Thus, a counter measure, increase in fish spill is in place. However, the increase in fish spill has a tradeoff in hydropower production as less water flows through the turbine. The U.S. provides additional benefits to Canada through the Canadian Entitlement, a payment equal to half of the expected additional hydropower generated due to cooperative management of the CRT dams. The collective monetary benefit from flood control and hydropower for among countries determine the utility of cooperation and non-cooperation (economic variables) for each country as described in Sect. 3.2.2. The social

preferences in different scenarios determine different values for utility of cooperation and non-cooperation depending on the actor's social preference. Thus, the directions of these relationships are conditional (Fig. 2). Having higher utility for cooperation under CRT results in a higher probability of cooperation. However, under changing social preferences if the utility of non-cooperation is higher, the probability of cooperation decreases. In sum, increase in cooperation for Canada results in decrease of dynamic storage threshold, Canada operates their reservoirs for downstream flood control, similarly increase in cooperation for the U.S. result in increase of the dynamic storage threshold, the U.S. operated for maximum hydropower generation, thus creating two similar feedback loops for Canada and the U.S. (Fig. 2).
* * *
2. I am puzzled about equation (3) and (4):

(1) The simplified reservoir operation rule indicated by equation (3) and (4) is used to determine the outflow, which is considered as vital factor in the model. It's suggested to cite corresponding references and add justification description for these equations.

(2) It's worth noting that $n_{CA}$ is an important parameter for outflow of Canada. What's the explicit connotation of $n_{CA}$ and how to determine it?

(3) The outflow is dominated by storage thresholds (i.e., $S_{CAthreshold}$ and $S_{USthreshold}$). The storage threshold is always between the target flood control storage ($S_{FCthreshold}$) and target hydropower storage ($S_{HPthreshold}$) as shown in Figure 3, as storage threshold is estimated by linearly aggregating $S_{FCthreshold}$ and $S_{HPthreshold}$ in equations (5) and (6), which is prone to simultaneously increase flood damage and decrease hydropower production. Please give more justification description.
* * *
*Authors' response*: Thank you for these comments and suggestions.

(1) The outflow from reservoirs are indeed the important variables, to which the variables for benefits, and feedback to the cooperation are dependent on. We have developed those sets of simplified reservoir operation equations (3-6) ourselves based on the conceptual understanding of the reservoir operation processes, including reviews of USACE (2003), to infer outflow from inflow and storage level. These three variables – storage, inflow, and outflow are presented in Section S1 of the supplementary material.

To simply describe the equations 3-4, they are a sequence of conditional statements. Here is a description for the U.S. outflow ($Q_{oUS}$). The first check is to examine whether the reservoir storage is full. In case the incoming volume of water with current level exceed the maximum capacity (Note, maximum capacity for U.S. is also their operating level to maximize hydropower), it should release incoming flow with addition of certain portion of the reservoir too. If the volume check is fine, the second check is whether the inflow is greater than the historical peak outflow that have occurred in the past (i.e., ~8000 m$^3$/s as shown in Fig. S3 (average monthly outflow)). In case its true then release should be the historical peak outflow. Otherwise if inflow is less than the historical peak flow then the release is inflow with addition of certain volume from the existing storage level.

Similarly, for the Canadian reservoir, the conditions are mostly similar to the U.S. If the first storage volume check is above the maximum capacity, there is additional second check if the inflow is higher than historical peak outflow, if true the release is that peak flow (i.e., ~2700 m³/s as shown in Fig. S9 (Mica monthly outflow)), otherwise release is inflow in addition of the certain volume from existing storage. However, if the first storage volume check is below the capacity, there is a second check which determine if the portion of the inflow (determined by $n_{CA}$) is greater than the historical peak, then release is only that peak flow. Otherwise, just release the portion of that inflow in addition to certain volume from the reservoir.

(2) Note that releasing only the portion of the inflow was necessary in order to prevent over release of water to make space for incoming flood flow later. That drawdown is not necessary over the year round. So, $0 < n_{CA} <= 1$ as a fraction ensure less water is released and stored in the reservoir when not necessary. The $n_{CA}$ is parameterized during calibration.

(3) You are correct the $S_{CAthreshold}$ and $S_{USthreshold}$ represent the current operation determined by the level of probability to cooperation. It is to be noted that $S_{HPthreshold}$ and $S_{FCthreshold}$ for the U.S. and Canada is different, and as we can see in equations 5 and 6. In the case when probability to cooperation is ~1 the second expression of the both equations tend to 0, and Canada operates its reservoir in full flood control mode, similarly the U.S. opt for maximum hydropower production mode. Also note that for flood control the reservoir should draw down to make space for oncoming flow, similarly for hydropower production the reservoir should be kept at full to achieve the maximum head. You are also correct the U.S. may need to operate their own dams to prevent flood damages downstream, while Canada can just produce their own hydroelectricity. This chaotic case only happens when there is no-cooperation, or only when conflict occurs. And this kind of behavior wouldn't happen in the current modelled scenarios. However, numerically that is possible as the system can model cooperation and conflict.

3. Please check whether the second '$C_{CA}$' is a typo in equation (6).

*Authors' response*: Thank you for pointing to this error. It is indeed $C_{US}$ in equation 6.

*Revision (#2.4):* The correction is made in line 410.

4. The motivation of applying logit dynamics functions to simulate the cooperation probability variables $C_{CA}$ and $C_{CA}$ should be detailed in line 378.

*Authors' response*: We will clarify the motivation for logit dynamics in the revised manuscript.

*Revision (#2.5):* The text was revised from lines through 473 – 486, to elaborate the motivation of using logit dynamics:

We chose to use logit dynamics (Hofbauer and Sigmund, 2003) over replicator dynamics (Taylon and Jonker, 1978) because the former enables us to incorporate actors' innate social preferences, i.e., each actor internally compares two choices (e.g., cooperation vs. defection) in

terms of net utilities that reflect their social preferences and then makes a probabilistic choice. In comparison, replicator dynamics are based on social comparisons of externally observable material payoffs and social imitation, i.e., each actor sees externally observable material payoffs of other actors following a particular strategy, compares that strategy's payoff to the material payoff of his or her current strategy, and then deterministically choose the better strategy. Because logit dynamics is more compatible with representation of social preferences and because of its stochastic best response nature, we chose logit dynamics. Eq. (12) and (13) represent the rate of change in the cooperation probability of the two state actors based on logit dynamics:

> 5. It's unjustified to determine the hydropower without considering water head in equation (20), despite that the simulated series can fit the observed series well. Moreover, the threshold water flow is directly selected as 400 m$^3$ /s, which needs more description.

*Authors' response*: Thank you for this question. The total hydropower benefits from the U.S. treaty dams is the sum of Grand Coulee and other five dams. Only the Grand Coulee dam is the storage hydropower dam where we established the relation between past hydropower production (as a response variable) vs. forebay level and outflow (as independent variables) (Fig. S14). Other dams do not have significant head and are mostly run of river type hydropower plants. For this we have established the relation between past hydropower produced by five dams (as a response variable) vs. outflow from Grand Coulee and weighting factor that considers the operations to meet environmental demands (as independent variables). This later relationship was not the linear, so we have separated the data into two halves (below and higher than 4000 m$^3$ /s) (Fig. S15). The 400 m$^3$ /s is a typo and correct is 4000 m$^3$ /s. We will correct this.

*Revision (#2.6):* The correction is made in equation 19 and 20, and results based on daily data are updated as discussed in *Revision (#2.2)*

> 6. It's suggested to add another section in Methodology to describe the feedback loops on basis of the dynamic equations in Section 3.2.

*Authors' response*: We will revise the Fig.2 and elaborate in the feedback links of the system dynamics.

*Revision (#2.7):* The revisions were made as discussed in the *Revision (#2.3)*.

> 7. In line 677, how to determine whether the stability is achieved?

*Authors' response*: Thank you for this comment. In the warmup period of the simulation, the initial value of cooperation changes without the repetition of particular pattern, after which the pattern of the dynamics is well observed. For clarification, as shown in Fig.7b the initial increase in probability took three-time (month) step in simulation. We will clarify this in the revised manuscript.

*Additional response:* Since using the daily time step the model is responsive compared to monthly time step simulation. Thus, this text is removed in the revised manuscript.

> 8. In Figure 7(b), the trajectories of probability to cooperate perform notable periodicity, which needs to be well accounted.

*Authors' response***:** The periodicity is due to the dynamics of the change in cooperation tied to the streamflow which has seasonal pattern. We will clarify this in the revised manuscript.

***Revision (#2.8):*** We have revised the text in lines 820 – 823:

... During each time steps the probability to cooperation changes as shown in equations 12 and 13. The periodicity in the probability to cooperation is due to the seasonality in the streamflow pattern. It is to be noted that for the key decisions regarding the reservoir operations, the peak amplitude is the deciding criteria.

---

## Referee Report (RR1)

The manuscript has been significantly improved, and I appreciate the authors' efforts to respond to reviewer comments. Specifically, the introduction is greatly improved by the inclusion of a comprehensive literature review and justifications for the selection of variables, particularly with regard to social preferences. I recommend that the manuscript be published as-is.

---

## Author Response (AR2)

Though the authors have made substantial revisions to the manuscript, I believe that my prior comment regarding reservoir aggregation in major issue hasn't been addressed. I think the authors should improve their Methodology, and further update the results.

Lumping reservoirs storage for water supply may be reasonable when only focusing on the total amount of water supply. However, it's quite unjustified to lump reservoirs storage when it comes to flood control. Neither the hydrological connections between reservoirs nor the spatial distribution of flood is considered, which will lead the overestimated local flood control capacity, and further severe loss. A simple example is detailed below:
There are three cascade reservoirs A, B, and C, from upstream to downstream. The storages for flood control of A, B, C are 100, 200, and 300, respectively. Lumping reservoir storage indicates that the flood control storage becomes 600. The 600-level flood can be regulated through reservoir operation when the flood comes from upstream side of reservoir A. However, as the 600-level flood comes from the river segment between reservoirs B and C, the flood will exceed the regulate capacity of the system, which cannot be identified by lumping reservoir storage. Therefore, it's highly-risked to lump reservoirs storage for flood control.

*Authors' response*: We thank you for your kind review of our paper, your valuable comments earlier, and your comment on our revised manuscript. We appreciate that you have further concerns about the lumped system and have addressed them below.

We would like to illustrate our lumped system in the model through Fig. A below. In lumped system of Canada there are three treaty dams namely, Mica, Arrow and Duncan with their respective storages as 24.7, 10.3, and 1.77 km$^3$, which are 67, 28, and 5% of the total installed storage respectively. The stream through Mica and Arrow Dams is the primary stream order of Columbia river and the one through Duncan is a small tributary. These three dams are combined to calculate flood control and hydropower. We did consider all the hydrological connections and segments that constitute the inflows for each lumped system, as shown in Fig. A. Similarly, in lumped system of the U.S. there is only one storage dam i.e., Grand Coulee (GCL) Dam and rest of the treaty dams are non-storage / run-of-river hydropower plants as shown below in Fig. A. Therefore, the storage is not lumped for the U.S., and reservoirs are only lumped for the hydropower estimation, which is appropriate given the run-of-river design. The schematic of the streamflow and dams is also presented in Fig. 1 of the revised manuscript; and the nature of treaty dams is also illustrated in Table. 1 of the revised manuscript.

Furthermore, in our model we also considered hydrological connection through the international boundary to the location at The Dalles where flood damage usually occurs. We considered the tributaries along the international boundary that contribute to inflow in lumped system in U.S. (or the GCL dam) along with the outflow from lumped Canadian system. As illustrated by the comparison of observed and simulated streamflow for GCL dam (Fig. 5c and Fig. 6c of the revised manuscript), our computation replicates streamflow patterns appropriately for the paper question and scope. In addition, the hydropower benefits (i.e., the response variable) that depends on the streamflow (the independent variable) included in the model was able to accurately reproduce hydropower benefits for the U.S. (Fig. 5d and Fig. 6d of the revised

manuscript). We are confident that hydrologically the lumped system is appropriate for the research question and scope.

The spatial distribution of the flooding is an important topic and is one of the major limitations of the lumped models. However, the spatial distribution of the flood is not relevant for the scope of this study because the historic flood damages occurred downstream of Columbia river at "The Dalles" and which was one of the key motivation for the CRT, thus in our study we considered flood control benefit by specifically considering flooding at The Dalles.

We have added the description of the lumped system throughout the section *3.2 Equations and parameters,* for Canadian and U.S. inflow (in lines 383 – 392 in revised manuscript), and for hydropower production is the lumped systems (in lines 551 – 556) of the revised manuscript. We realize that elaborating on the lumped systems might clear potential confusion and help readers. Thus, we added a section 4 "*Lumped systems in the model*" in the supplemental material with the description shown in Fig. A below. We also elaborated text in the section "*3.1 Socio-hydrological system dynamics model*" of the re-revised manuscript as discussed below in *Revision*.

[Figure]

Figure A. Lumped systems in Canada and U.S.

*Revision:* We have added text (as shown by blue color) in third paragraph of the section "*3.1 Socio-hydrological system dynamics model*" as below:

The storage capacity of Canada (upstream) and the U.S. (downstream) are two important state (hydrological) variables which represent the aggregated storage of the treaty dams (Fig. 2), the operation of which is determined by the storage thresholds. The increase in a storage threshold results in an increase in the storage level. Three Canadian dams namely Mica, Duncan and Keenleyside are lumped into a single storage as all three dams are multifunctional for flood control and hydropower production. However, it should also be noted that Mica and Arrow Dams are the major dams in Canada contributing to flood control as those are along the primary stream order of Columbia River and Duncan Dam is in the small tributary (Fig. 1). In terms of storage volume Mica, Arrow and Duncan Dams are 24.7 km$^3$, 10.3 km$^3$, and 1.77 km$^{3,}$ or 67%, 28%, and 5% of total storage, respectively (Table 1). In the U.S., the Grand Coulee dam is the only multifunctional dam with useable storage for flood control. Given that the Grand Coulee is the only dam with storage in in the U.S. the system, we have only lumped the reservoirs for hydropower generation, not flood control. We used the lumped reservoir approach to simplify the system process required to investigate our research questions. The lumped approach is particularly appropriate because all the treaty dams work in coordination to achieve either of the hydropower benefits (by U.S. dams) or flood control (by Canadian dams). The schematic of the lumped system is also shown in Fig. S18, Section S4 of the supplemental material. In lumping the system, we have considered external input variables such as tributaries and added to the outflow from Canadian reservoir, or inflow to the U.S. reservoir. These dams along the Columbia River either have significant flood control capacity or significant hydropower production capacity (Table 1). Thus, the simplified reservoir operation described below in Sect. 3.2.1 was implemented in the lumped storages on each side of the border, which represent collective operation of all the treaty dams within each country. Other hydrological variables in the model (i.e., flows in the CL diagram) are inflow into Canadian storage, outflow from Canadian storage plus intermediate tributaries, inflow into the U.S. storage, and outflow from the U.S. storage. The higher the outflow from the dams, the lower the flood control as flood damages increase. A portion of the reservoir outflow passes through hydroelectric turbines, thus more outflow yields higher hydropower benefit.